# FAKE-IN-FACEXT: TOWARDS FINE-GRAINED EXPLAINABLE DEEPFAKE ANALYSIS

## ABSTRACT

The advancement of Multimodal Large Language Models (MLLMs) has bridged the gap between vision and language tasks, enabling the implementation of Explainable DeepFake Analysis (XDFA). However, current methods suffer from a lack of fine-grained awareness: the description of artifacts in data annotation is unreliable and coarse-grained, and the models fail to support the output of connections between textual forgery explanations and the visual evidence of artifacts, as well as the input of queries for arbitrary facial regions. As a result, their responses are not sufficiently grounded in Face Visual Context (Facext). To address this limitation, we propose the **Fake-in-Facext (FiFa)** framework, with contributions focusing on data annotation and model construction. We first define a Facial Image Concept Tree (FICT) to divide facial images into fine-grained regional concepts, thereby obtaining a more reliable data annotation pipeline, FiFa-Annotator, for forgery explanation. Based on this dedicated data annotation, we introduce a novel Artifact-Grounding Explanation (AGE) task, which generates textual forgery explanations interleaved with segmentation masks of manipulated artifacts. We propose a unified multi-task learning architecture, FiFa-MLLM, to simultaneously support abundant multimodal inputs and outputs for fine-grained Explainable DeepFake Analysis. With multiple auxiliary supervision tasks, FiFa-MLLM can outperform strong baselines on the AGE task and achieve SOTA performance on existing XDFA datasets. The code and data will be made open-source.

## 1 INTRODUCTION

Artificial Intelligence Generated Content (AIGC) blurs the boundary between fiction and reality. Unauthorized DeepFake images can be exploited to disseminate misinformation, potentially leading to significant social issues and security threats (Tolosana et al., 2020; Turton & Martin, 2020). Deep-Fake Analysis (DFA) thus becomes crucial for regulating technological applications and mitigating associated societal risks. Over the past two years, the rapid advancement of Multimodal Large Language Models (MLLMs) (Liu et al., 2023) has bridged the gap between vision and language tasks. This progress has enabled the research to move beyond simple binary classification, expanding its scope to generative visual question answering. Many works for Explainable DeepFake Analysis (XDFA) (Zhang et al., 2024b; Huang et al., 2024; Qin et al., 2025; Xu et al., 2025; Sun et al., 2025; Guo et al., 2025) have emerged.

However, existing MLLMs for XDFA still have a fundamental limitation: they suffer from a lack of fine-grained awareness. Specifically, there are the following issues: (1) in terms of data annotation, existing methods over-rely on GPTs to identify artifact-existing regions while neglecting the application of prior knowledge, leading to unreliability in artifact explanations. Furthermore, the set of concepts used to describe the locations of artifacts is relatively small and only includes a limited number of concepts, which results in imprecise descriptions of artifact locations. For example, an artifact occurring on "the left nasal ala" can only be described as appearing in "the region around the nose". (2) In terms of model construction, these models only output textual explanations and lack the connection between textual forgery explanations and the visual evidence of artifacts. Additionally, they do not support flexible linguistic or visual prompts, thus failing to provide users with targeted discussions regarding facial regions of interest. To address the aforementioned issues, we

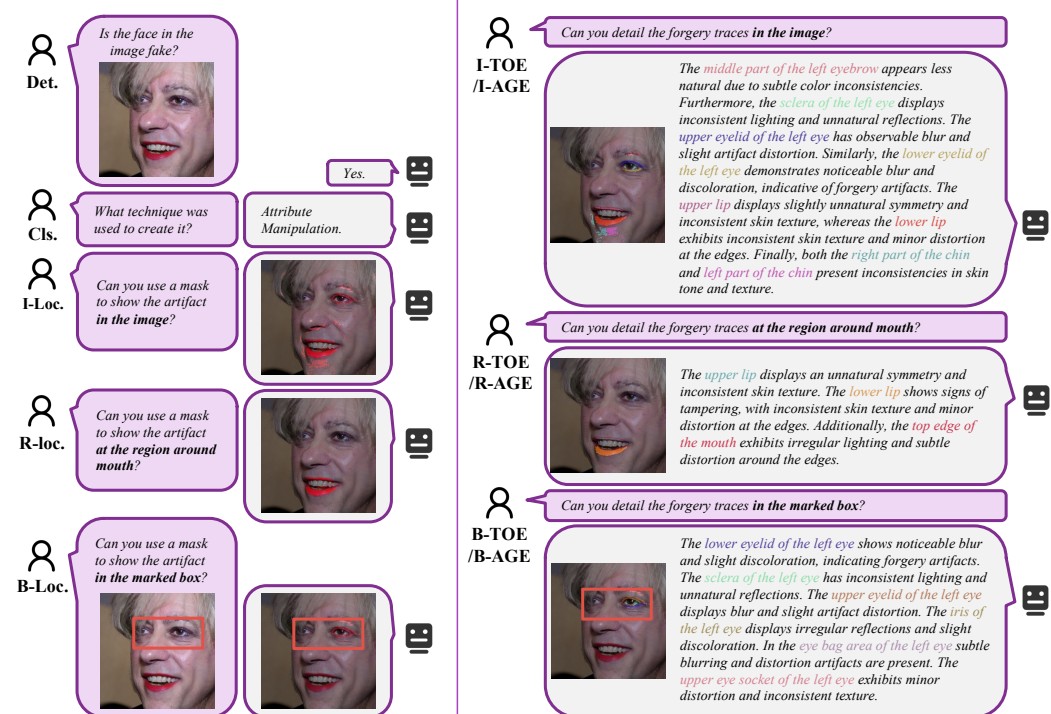

Figure 1: 11 Tasks in **FiFa-11**: Detection (Det.), Classification (Cls.), Image-Level Localization (I-Loc.), Region-Level Localization (R-Loc.), Box-Level Localization (B-Loc.), Image-Level Text-Only Explanation (I-TOE), Region-Level Text-Only Explanation (R-TOE), Box-Level Text-Only Explanation (B-TOE), Image-Level Artifact-Grounding Explanation (I-AGE), Region-Level Artifact-Grounding Explanation (R-AGE), Box-Level Artifact-Grounding Explanation (B-AGE).

propose the **Fake-in-Facext (FiFa)** framework, which aims to ensure that the responses of MLLMs for XDFA are fully grounded in Face Visual Context (Facext).

For more reliable data annotation, we propose an automated data annotation pipeline named **FiFa-Annotator**. We introduce the Facial Image Concept Tree (FICT), which comprises 8 hierarchical levels to model the context of facial images. The FICT includes 112 Atomic Concepts and 72 Parent Concepts, which are utilized for the fine-grained division of facial images. First, the list of Atomic Concepts containing artifacts is determined based on the artifact coverage ratio within the corresponding facial region. Subsequently, a powerful MLLM (OpenAI, 2024) generates a detailed forgery explanation for each artifact-containing Atomic Concept. Finally, a robust LLM (OpenAI, 2023) aggregates these explanations derived from Atomic Concepts to synthesize comprehensive forgery explanations for Parent Concepts across different hierarchical levels. By leveraging prior knowledge, we obtain forgery explanation annotations with fewer hallucinations in comparison to existing automated annotation pipelines. Meanwhile, as we provide a more fine-grained set of concepts, the description of artifact locations is also more precise.

Existing MLLMs for XDFA are capable of performing tasks such as DeepFake Detection (Det.), Classification (Cls.), Localization (Loc.), and Text-Only Explanation (TOE). Within our FiFa framework, based on the customized FiFa-Annotator, we further extend the capabilities of XDFA: (1) Output pixel grounding capability. We introduce a novel task of Artifact-Grounding Explanation (AGE). The AGE task aims to output a natural language response explaining the forgery while simultaneously providing a segmentation mask that pinpoints the artifacts mentioned in the text. (2) Support for more flexible input queries. For the Loc., TOE, and AGE tasks, in addition to supporting queries about the entire face (Image-Level), we also enable the specification of facial regions of interest via textual prompts (Region-Level) and bounding box visual prompts (Box-Level). Consequently, we define a comprehensive set of 11 tasks for fine-grained XDFA, called **FiFa-11**. Samples are depicted in Figure 1. Using FiFa-Annotator, we have constructed the **FiFa-Instruct-1M** training

Table 1: Comparison of capabilities supported by existing work and our FiFa framework.

| Model | Dataset | QA-Pairs | Det. | Cls. | I-Loc. | I-TOE | I-AGE | R-Loc. | R-TOE | R-AGE | B-Loc./B-TOE/B-AGE |
|---|---|---|---|---|---|---|---|---|---|---|---|
| DDVQA-BLIP | DD-VQA | 15K | ✓ | | | ✓ | | | ✓ | | |
| FFAA | FFA-VQA | 20K | ✓ | ✓ | | ✓ | | | | | |
| DFA-GPT | DFA-Instruct | 127K | ✓ | ✓ | | ✓ | | | ✓ | | |
| FakeShield | MMTD-SET | 17K (For Face Forgery) | ✓ | | ✓ | ✓ | | | | | |
| **FiFa-MLLM (Ours)** | **FiFa-Instruct-1M (Ours)** | **1.38M** | ✓ | ✓ | ✓ | ✓ | ✓ | ✓ | ✓ | ✓ | ✓ |

dataset (containing 1.38 million samples) as well as an evaluation benchmark named **FiFa-Bench** for FiFa-11.

To address these challenging tasks in **FiFa-11**, we have constructed a unified multi-task learning architecture, FiFa-MLLM. Current MLLMs (Lai et al., 2024; Rasheed et al., 2024) typically introduce additional visual encoders dedicated to segmentation for pixel grounding capability, resulting in inefficient architectures. To address this, our **FiFA-MLLM** employs only one global visual encoder. This encoder concurrently generates suitable visual features for both LLM input and mask prediction. Furthermore, we propose a Multi-Task Decoder. By introducing distinct task-specific query embeddings, this decoder can simultaneously handle Artifact Mask Prediction and multiple auxiliary supervision tasks. Experiments reveal that auxiliary supervision of Region Mask Prediction effectively enhances the accuracy of Artifact Mask Prediction in both Loc. and AGE tasks. We also support bounding box visual prompts by introducing a Box Encoder. Consequently, the resulting MLLM demonstrates compelling advantages: despite having 0.94B fewer parameters than the strong baseline GLaMM (Rasheed et al., 2024), it achieves significantly superior performance on nearly all tasks in FiFa-11. On the existing XDFA test benchmarks, DD-VQA and DFA-Bench, FiFA-MLLM also achieves SOTA results.

Our contribution of the **Fake-in-Facext (FiFa)** framework for fine-grained Explainable DeepFake Analysis can be summarized as follows:

1. We define a comprehensive task set, **FiFa-11**, to pioneerly ground the responses of XDFA-aimed MLLMs in the Face Visual Context.

2. We propose a novel data annotation pipeline, **FiFa-Annotator**, and construct the **FiFa-Instruct-1M** training dataset alongside the **FiFa-Bench** evaluation dataset with it. FiFa-Instruct-1M is the largest training dataset for the XDFA field currently known.

3. We propose **FiFa-MLLM**, the first XDFA-aimed model capable of supporting Artifact-Grounding Explanation and responding to bounding box visual prompts.

4. Through well-designed architecture, our FiFa-MLLM significantly outperforms the strong baseline across almost all tasks in FiFa-11.

## 2 RELATED WORKS

**Explainable DeepFake Analysis.** The rapid advancement of MLLMs has expanded the application scope of DFA beyond mere detection and classification. Models such as DDVQA-BLIP (Zhang et al., 2024b), FFAA (Huang et al., 2024), DFA-GPT (Qin et al., 2025) have demonstrated textual explanation capabilities, while FakeShield (Xu et al., 2025) has further integrated localization. Table 1 contrasts the capabilities supported by existing works with our proposed framework. Training XDFA-aimed MLLMs necessitates high-quality image-text annotations. Existing data annotation pipelines can be broadly classified into two categories: (1) Manual annotation (Zhang et al., 2024b). This method relies on human annotators to identify artifact-existing facial regions and select appropriate descriptive items from predefined analytical options for each region. While providing reliable forgery explanations, this method suffers from scalability limitations and often yields monotonous textual descriptions. (2) Automated annotation (Huang et al., 2024; Qin et al., 2025; Xu et al., 2025). This method employs meticulously crafted prompts to leverage powerful MLLMs like and GPT-4o (OpenAI, 2024) for generating holistic facial forgery explanations. However, this method heavily depends on the forgery understanding abilities of the underlying MLLMs, thus reducing the reliability of textual explanations (Sun et al., 2025).

**Pixel Grounding and Box-Level Understanding in MLLM.** Seminal works, including GPT4RoI (Zhang et al., 2024a), Kosmos-2 (Peng et al., 2023), and Shikra (Chen et al., 2023), pioneer the Box-Level understanding capability in MLLMs, enabling conversational focus specification through

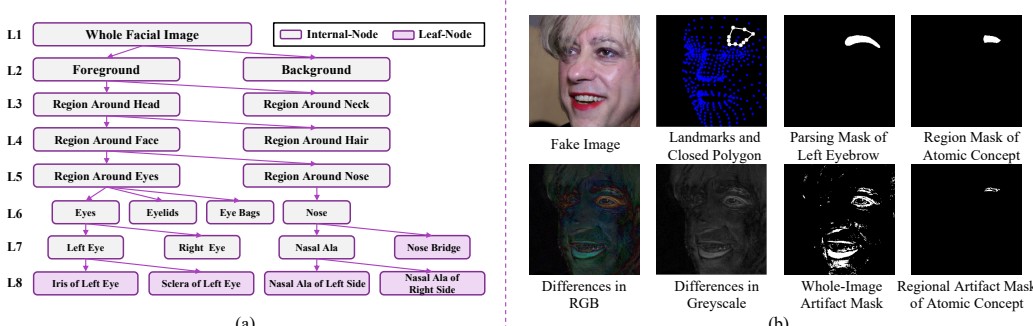

Figure 2: (a) Facial Image Concept Tree (FICT). We have selected part of the nodes for illustration. For convenience, some internal-nodes do not have their child-nodes drawn. For the complete list of nodes, refer to Appendix A.1. (b) Image processing procedure in FiFa-Annotator.

bounding box visual prompts. Pixel grounding is first introduced into MLLMs by LISA (Lai et al., 2024) and GLaMM (Rasheed et al., 2024). These approaches not only employ the CLIP (Radford et al., 2021) encoder to provide visual features for LLM inputs but also necessitate integrating the SAM (Kirillov et al., 2023) encoder specifically for mask prediction.

**Transformer Decoder in Computer Vision.** The success of DETR (Carion et al., 2020) in object detection has motivated researchers to employ the transformer decoder for solving computer vision problems. MaskFormer (Cheng et al., 2021) proposed a unified approach for semantic and instance-level segmentation tasks, where each segment is represented by a query in the transformer decoder. In Face Perception, Faceptor (Qin et al., 2024), Facexformer (Narayan et al., 2024), and Q-Face (Sun et al., 2024) utilize task-specific query embeddings to simultaneously handle multiple tasks, such as face recognition, facial attribute understanding, and face parsing. In our proposed FiFa-MLLM, we implement a two-way transformer decoder having a group of task-specific query embeddings. It serves as the Multi-Task Decoder to concurrently address the primary task of Artifact Mask Prediction alongside a set of auxiliary supervision tasks.

## 3 DATA ANNOTATION PIPELINE

In this section, we begin by describing the proposed Facial Image Concept Tree (FICT). Building upon this hierarchical structure, we then construct our novel data annotation pipeline, FiFa-Annotator. Finally, we conclude the statistics for the generated large-scale training dataset FiFa-Instruct-1M and the evaluation benchmark FiFa-Bench. FiFa-Annotator greatly expands the vocabulary for describing artifact locations, enabling more fine-grained descriptions. Meanwhile, through the identification of Artifact-Existing Concepts, it enhances the reliability of data annotation.

### 3.1 FACIAL IMAGE CONCEPT TREE

To model context in facial images, we propose the Facial Image Concept Tree (FICT), depicted in Figure 2(a). This hierarchical tree comprises 8 levels, where each node represents a region concept within the facial image. We designate concepts on leaf-nodes as Atomic Concepts (totaling 112) and concepts on internal-nodes as Parent Concepts (totaling 72). The concept of the root node represents the "whole facial image," which is the spatially largest Parent Concept.

Leveraging existing facial landmark localization (Lugaresi et al., 2019) and face parsing (Zheng et al., 2022a) tools, we can derive a Region Mask corresponding to each Atomic Concept for a given facial image. The first row of Figure 2(b) illustrates the Region Mask acquisition process for the "middle part of the left eyebrow" concept: the intersection is computed between a closed polygon formed by specific landmarks and the left eyebrow mask generated by face parsing. Region Masks for other Atomic Concepts can be similarly produced through analogous predefined procedures. Region Masks for Parent Concepts are subsequently obtained by performing the union operation on the masks of their constituent Atomic Concepts.

Table 2: Statistics of FiFa-Instruct-1M and FiFa-Bench.

| Task | Data Source | Training | Dev. | Test |
|---|---|---|---|---|
| Det./Cls. | FFHQ/CelebA/DFFD | 160000 | 15439 | 16000 |
| I-Loc. | | 10000 | 439 | 1000 |
| R-Loc. | | 300791 | 1000 | 1000 |
| B-Loc. | | 182170 | 1000 | 1000 |
| I-TOE | | 9940 | 438 | 992 |
| R-TOE | DFFD: FaceAPP | 300744 | 1000 | 1000 |
| B-TOE | | 182097 | 1000 | 1000 |
| I-AGE | | 9796 | 431 | 976 |
| R-AGE | | 124113 | 1000 | 1000 |
| B-AGE | | 103561 | 1000 | 1000 |
| Total | | 1383212 | 22747 | 24968 |

## 3.2 FIFA-ANNOTATOR

Building upon the FICT, we construct a novel annotation pipeline termed FiFa-Annotator. The detailed procedure for generating data for the AGE task is outlined below. Data for the Loc. and TOE are concurrently generated.

**Step 1: Whole-Image Artifact Mask Extraction.** We compute per-pixel differences in RGB channels between real and manipulated facial images, converting the result to grayscale. A binary Whole-Image Artifact Mask is generated by thresholding the top 5% of intensity values (see the second row of Figure 2(b)). What deserves special mention is that we only use facial images forged via attribute manipulation techniques to produce data for FiFa. This is because other forgery methods (e.g., identity/expression swapping) induce maximal pixel changes unrelated to artifacts, whereas attribute manipulation concentrates unnatural traces in altered regions, enabling reliable artifact localization through the per-pixel difference operation.

**Step 2: Identification of Artifact-Existing Concepts.** Leveraging the Region Mask derived in Section 3.1, we can compute the Regional Artifact Mask for each concept via intersection with the Whole-Image Artifact Mask (see the second row of Figure 2(b)). The artifact coverage ratio is defined as the proportion of artifact pixels within each concept's Region Mask. Atomic/Parent Concepts are flagged as artifact-existing if they simultaneously satisfy: (1) Rank $\leq X$ in artifact coverage ratio descending order. (2) Artifact coverage ratio $\geq Y\%$. (3) Number of artifact pixels $\geq Z$. In our implementation, $X$, $Y$, and $Z$ are set to 50, 10, and 50, respectively. The concepts of "whole facial image", "foreground", "region around head", and "region around face" are included in the artifact-existing Parent Concepts by default.

**Step 3: Forgery Explanation Generation for Atomic Concepts.** For each identified artifact-existing Atomic Concept, GPT-4o (OpenAI, 2024) synthesizes detailed forgery descriptions using well-designed prompts (refer to Appendix A.3). Our prompt refers to forgery analysis perspectives adapted from prior work (Xu et al., 2025).

**Step 4: Forgery Explanation Synthesis for Parent Concepts.** Following hierarchical relationships in FICT, ChatGPT (OpenAI, 2023) aggregates forgery explanations for Atomic Concepts to generate coherent forgery explanations for artifact-existing Parent Concepts, with well-designed prompts (refer to Appendix A.3). In this way, each Atomic Concept referenced in explanations for Parent Concepts has a precomputed Regional Artifact Mask, enabling I-AGE/R-AGE tasks. We further compute bounding boxes for Region Masks of Parent Concepts to support B-AGE tasks.

**Step 5: B-AGE Data Augmentation.** To enhance fine-grained forgery analysis, we augment B-AGE data by: (1) Randomly generating 20 bounding boxes per image. (2) Retaining boxes encompassing $\geq 3$ artifact-existing Atomic Concepts. (3) Synthesizing Box-Level forgery explanations using a prompt similar to that in Step 4.

## 3.3 STATISTICS OF FIFA-INSTRUCT-1M & FIFA-BENCH

We used FiFa-Annotator to generate data covering the FiFa-11 task set using the source data from DFFD (Dang et al., 2020). The training data is named FiFa-Instruct-1M, including 1.38M QA-pairs. This is the largest-scale training dataset for Explainable DeepFake Analysis field currently known. The test data is named FiFa-Bench, including a development set and a test set. Table 2 gives the

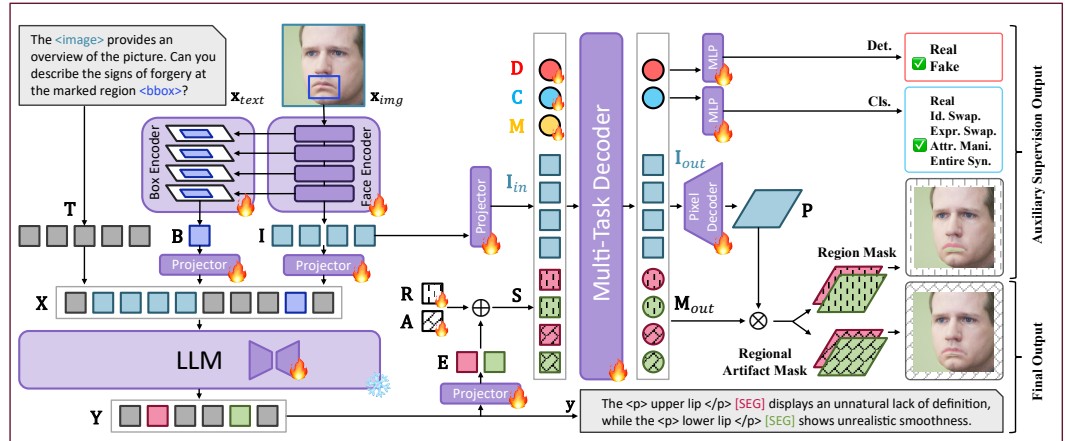

Figure 3: Overall architecture for the proposed FiFa-MLLM.

statistics about the constructed data. In order to enable the model to detect different categories of forgery techniques, we use samples from DFFD for forgery images in Dec. and Cls. tasks, covering four types of forgery techniques: identity swapping, expression swapping, attribute manipulation, and entire face synthesis.

## 4 FIFA-MLLM

In this section, we first describe the model architecture of our proposed FiFa-MLLM and then specifically explain the auxiliary supervision of Region Mask Prediction. FiFa-MLLM greatly improves the fine-grained awareness by supporting multimodal outputs and inputs—where multimodal outputs enable connections between textual forgery explanations and the visual evidence of artifacts, and multimodal inputs allow queries about arbitrary facial regions.

### 4.1 MODEL ARCHITECTURE

The proposed FiFa-MLLM (as shown in Figure 3) comprises five core components: (1) Face Encoder, (2) Box Encoder, (3) Large Language Model (LLM), (4) Multi-Task Decoder, and (5) Pixel Decoder. Several projectors are incorporated to transform features across different representation spaces. Our model is meticulously designed to process both textual and optional visual prompts (bounding boxes), allowing for interaction at multiple levels of granularity and generating Artifact-Grounding text responses.

**Facial Image Encoding.** To achieve robust facial image encoding, we utilize a 12-layer ViT-B (Dosovitskiy et al., 2021) as the face encoder, pre-trained via the FaRL (Zheng et al., 2022b) framework. Given an input facial image $\mathbf{x}_{img}$, the visual encoder produces:

$$\mathbf{I} = \text{FaceEncoder}(\mathbf{x}_{img}) \in \mathbb{R}^{L_I \times D_I}. \quad (1)$$

Subsequently, $\mathbf{I}$ is projected into the natural language space, yielding $\mathbf{I}' \in \mathbb{R}^{L_I \times D_{llm}}$.

**Bounding Box Visual Prompt Encoding.** To enhance fine-grained facial context understanding and support bounding box visual prompts, we introduce the Box Encoder. This module constructs a hierarchical feature pyramid from four selected face encoder layers, then employs RoIAlign (He et al., 2017) to generate a feature map. Aggregating these features produces a unified representation:

$$\mathbf{B} = \text{BoxEncoder}(\mathbf{x}_{img}, bbox) \in \mathbb{R}^{1 \times D_B}, \quad (2)$$

which is then projected to the natural language space as $\mathbf{B}' \in \mathbb{R}^{1 \times D_{llm}}$.

**Interleaved Vision-Language Sequence Processing.** The input text prompt $\mathbf{x}_{text}$ is tokenized into $\mathbf{T} \in \mathbb{R}^{L_T \times D_{llm}}$. Tokens corresponding to special vocabularies "<image>" and "<bbox>" in $\mathbf{T}$ are replaced with $\mathbf{I}'$ and $\mathbf{B}'$ respectively, forming an interleaved sequence: $\mathbf{X} \in \mathbb{R}^{L_X \times D_{llm}}$. The LLM generates the output token sequence:

$$\mathbf{Y} = \text{LLM}(\mathbf{X}) \in \mathbb{R}^{L_Y \times D_{llm}}, \quad (3)$$

Table 3: The first and third rows compare our proposed FiFa-MLLM with the strong baseline GLaMM in a multi-task setting. The second and third rows compare the performance of FiFa-MLLM in single-task and multi-task settings.

| Method | | Det. | | Cls. | I-TOE | I-Loc. | I-AGE | |
| | ACER ↓ | Acc. ↑ | F1 ↑ | Acc. ↑ | METEOR ↑ | mIoU ↑ | METEOR ↑ | mIoU ↑ |
|---|---|---|---|---|---|---|---|---|
| GLaMM (Multi-Task) | 6.97 | 93.03 | 92.90 | 90.48 | 21.3 | 20.1 | 21.4 | 22.5 |
| FiFa-MLLM (Single-Task) | 16.59 | 83.41 | 83.75 | 74.18 | 20.9 | 25.6 | 21.2 | 23.7 |
| **FiFa-MLLM (Multi-Task)** | **4.47** | **95.53** | **95.59** | **93.00** | **23.0** | **31.6** | **23.0** | **29.6** |
| Method | R-TOE | R-Loc. | R-AGE | | B-TOE | B-Loc. | B-AGE | |
| | METEOR ↑ | mIoU ↑ | METEOR ↑ | mIoU ↑ | METEOR ↑ | mIoU ↑ | METEOR ↑ | mIoU ↑ |
| GLaMM (Multi-Task) | 18.7 | 24.9 | 20.7 | 25.0 | 21.2 | 24.3 | 20.5 | 27.2 |
| FiFa-MLLM (Single-Task) | 18.5 | 23.0 | 21.0 | 23.9 | 19.1 | 24.5 | 20.1 | 24.7 |
| **FiFa-MLLM (Multi-Task)** | **19.7** | **30.6** | **21.5** | **31.0** | **21.5** | **30.4** | 20.3 | **30.7** |

Table 4: Comparison on DD-VQA.

| Method | Det. | | | | I-TOE | | | |
| | Acc. ↑ | Recall ↑ | Precision ↑ | F1 ↑ | BLEU-4 ↑ | CIDEr ↑ | ROUGE_L ↑ | METEOR ↑ |
|---|---|---|---|---|---|---|---|---|
| DDVQA-BLIP-TI | 87.49 | 93.41 | 86.97 | 90.07 | 40.8 | 2.057 | 60.9 | 34.6 |
| **FiFa-MLLM (Ours)** | **88.76** | **95.24** | **87.72** | **91.32** | **48.1** | **2.869** | **65.4** | **40.4** |

which is then decoded into human-readable text $\mathbf{y}$.

**Multi-Task Processing.** The Multi-Task Decoder adopts a two-way transformer architecture (Kirillov et al., 2023). It can process two types of tokens: image tokens and task-specific query tokens:

$$\mathbf{I}_{out}, \mathbf{T}_{out} = \text{MultiTaskDecoder}(\mathbf{I}_{in}, \mathbf{T}_{in}). \tag{4}$$

Here, $\mathbf{I}_{in} \in \mathbb{R}^{L_I \times D_{de}}$ is derived by projecting $\mathbf{I}$. During training, different embeddings are selected as $\mathbf{T}_{in}$ based on the task that the current training sample belongs to. For Det., $\mathbf{T}_{in}$ is set to the detection embedding $\mathbf{D} \in \mathbb{R}^{1 \times D_{de}}$; for Cls., $\mathbf{T}_{in}$ is set to the classification embedding $\mathbf{C} \in \mathbb{R}^{1 \times D_{de}}$; for Loc. or AGE, $\mathbf{T}_{in}$ is set to the mask embedding $\mathbf{M} \in \mathbb{R}^{1 \times D_{de}}$ and the semantic embedding $\mathbf{S} \in \mathbb{R}^{1 \times D_{de}}$. $\mathbf{D}$, $\mathbf{C}$, and $\mathbf{M}$ are all randomly initialized for training.

**Output Decoding.** For the sample of the Det. and Cls. tasks, $\mathbf{T}_{out} \in \mathbb{R}^{1 \times D_{de}}$ is fed into a 2-layer MLP to obtain predictions. For the sample of the Loc. and AGE tasks, there are some special vocabularies in the output text $\mathbf{y}$: "<p>" and "</p>" denote the starting and ending points of Atomic Concepts in the forgery explanations, while "[SEG]" indicates where an Artifact Mask should be output. The output token in $\mathbf{Y}$ corresponding to "[SEG]" is projected as Concept Embedding $\mathbf{E} \in \mathbb{R}^{1 \times D_{de}}$ and we use $\mathbf{E}$ as the semantic embedding $\mathbf{S}$ in this section. The output of image tokens $\mathbf{I}_{out}$ is reshaped and upsampled by the Pixel Decoder, obtaining:

$$\mathbf{P} = \text{PixelDecoder}(\mathbf{I}_{out}) \in \mathbb{R}^{D_{de} \times H \times W}. \tag{5}$$

The Regional Artifact Mask for an Atomic Concept is computed via the dot product between $\mathbf{P}$ and the corresponding output of the mask embedding, $\mathbf{M}_{out}$. For simplicity, we have formalized the Mask Prediction process assuming the output text $\mathbf{y}$ contains only one occurrence of "[SEG]". In practice, the input $\mathbf{I}_{in}$ is duplicated based on the number of times "[SEG]" appears to generate the masks for all Atomic Concepts simultaneously.

Notably, during the inference stage, FiFa-MLLM's final outputs only include LLM-generated text $\mathbf{y}$ for all tasks and the Regional Artifact Mask for Loc. and AGE tasks. The auxiliary supervision of Det. and Cls. is aimed at enhancing the Face Encoder's forgery understanding capability.

## 4.2 AUXILIARY SUPERVISION OF REGION MASK PREDICTION

In Section 4.1, we predict the Regional Artifact Mask for each Atomic Concept via Concept Embedding $\mathbf{E}$. As detailed in Section 3.2's data annotation pipeline, Region Masks are acquired alongside Regional Artifact Masks for each Atomic Concept. In this section, we incorporate Region Mask Prediction as an auxiliary supervision to improve the accuracy of the Artifact Mask Prediction and enhance the fine-grained Facext understanding. To implement this, we introduce randomly initialized Region Embedding $\mathbf{R} \in \mathbb{R}^{1 \times D_{de}}$ and Artifact Embedding $\mathbf{A} \in \mathbb{R}^{1 \times D_{de}}$. The Concept Embedding corresponding to each Atomic Concept is combined with both the Region Embedding and Artifact Embedding, forming two distinct Semantic Embeddings. This enables the simultaneous generation of both a Region Mask and a Regional Artifact Mask for every Atomic Concept. The efficacy of this auxiliary supervision is empirically validated via ablation studies in Section 5.3.

Table 5: Comparison on DFA-Bench.

| Method | Det. ACER ↓ | Cls. Acc. ↑ | I-TOE ROUGE-L ↑ |
|---|---|---|---|
| DFA-GPT | 5.04 | 92.74 | 42.54 |
| **FiFa-MLLM (Ours)** | **4.54** | **92.95** | **42.78** |

Table 6: Model Design Ablation for FiFa-MLLM. For TOE, Loc., and AGE Tasks, we report the mean performance of the Image-Level, Region-Level, and Box-Level.

| | | GLaMM | FiFa-MLLM Baseline 1 | FiFa-MLLM Baseline 2 | FiFa-MLLM |
|---|---|---|---|---|---|
| Vision Encoder | Unified | ✗ | ✓ | ✓ | ✓ |
| | Face Pre-trained | ✗ | ✓ | ✓ | ✓ |
| | Learnable | ✗ | ✗ | ✓ | ✓ |
| Auxiliary Supervision | Det. and Cls. | ✗ | ✗ | ✓ | ✓ |
| | Region Mask | ✗ | ✗ | ✗ | ✓ |
| Det. | ACER ↓ | 6.97 | 5.21 | 4.79 | **4.47** |
| Cls. | ACC ↑ | 90.48 | 90.74 | 92.72 | **93.00** |
| TOE | METEOR ↑ | 20.4 | 19.8 | 21.3 | **21.4** |
| Loc. | mIoU ↑ | 23.1 | 24.6 | 30.0 | **30.9** |
| AGE | METEOR ↑ | 20.9 | 20.4 | **21.7** | 21.6 |
| | mIoU ↑ | 24.9 | 21.0 | 30.0 | **30.4** |

## 5 EXPERIMENTS

### 5.1 IMPLEMENTATION DETAILS

We implement FiFa-MLLM using PyTorch. The Face Encoder is initialized from a ViT-B model pre-trained with the FaRL (Zheng et al., 2022b) framework. The Box Encoder adopts the RoIAlign (He et al., 2017) structure. For the LLM, we utilize the Vicuna (Zheng et al., 2023) LLM with 7B parameters, initialized from the LLM weights of GLaMM-GranD-Pretrained (Rasheed et al., 2024). The Multi-Task Decoder employs the lightweight two-way transformer decoder proposed in SAM (Kirillov et al., 2023). The Pixel Decoder follows Faceptor (Qin et al., 2024) and is configured with two consecutive $2 \times 2$ deconvolutional layers. Projectors are implemented as two-layer MLPs using GELU activation functions. For the experimental hyperparameters of the training process, please refer to Appendix B.

### 5.2 PERFORMANCE EVALUATION

We report the performance of the multi-task models on the test set of FiFa-Bench in Table 3. For a strong baseline, we first fine-tuned GLaMM with identical hyperparameters. Our proposed FiFa-MLLM achieves significantly superior performance over GLaMM on nearly all evaluation metrics. We further compare FiFa-MLLM's performance under multi-task and single-task settings. To ensure fair comparison, all single-task models are trained with sample amounts identical to their corresponding subsets in the multi-task setting. Results demonstrate that multi-task learning effectively enhances data efficiency, yielding better-performing models while utilizing equivalent training data. We also evaluate the FiFa-MLLM on existing XDFA datasets: DD-VQA (Zhang et al., 2024b) and DFA-Bench (Qin et al., 2025), as shown in Tables 4 and 5. We use the same parameter initialization as in section 5.1, and train on the DD-VQA training set and DFA-Instruct respectively. Our model achieves competitive performance across both benchmarks, with particularly notable gains on the I-TOE task of DD-VQA. These quantitative results substantiate the effectiveness of our architecture design. Refer to Appendix C for qualitative results.

### 5.3 ABLATION STUDIES

**Model Design Ablation for FiFa-MLLM.** We conduct ablation experiments on the model design in Table 6. In FiFa-MLLM Baseline 1, we employ a unified Face Encoder design by replacing the CLIP-ViT-H/14 and SAM-ViT-H/16 in GLaMM (Rasheed et al., 2024) with a parameter-frozen FaRL-ViT-B/16. While this model baseline demonstrates performance comparable to GLaMM with advantages on certain tasks and disadvantages on others, it achieves a significant reduction in parameter overhead by 0.94B.

Table 7: Data Ablation on FiFa-11.

| Training Data | Det. (Intra-Domain) ACER ↓ | Det. (Cross-Domain) ACER ↓ |
|---|---|---|
| Det. | 0.90 | 29.32 |
| Det.-TOE | 0.70 | 29.08 |
| Det.-TOE-Loc. | 0.35 | 28.79 |
| Det.-I | 0.61 | 28.94 |
| Det.-I-R | 0.33 | 28.89 |
| Det.-TOE-L-AGE /Det.-I-R-B | **0.30** | **28.57** |

Table 8: Superiority of FiFa-Annotator.

| I-TOE Data | ACER ↓ |
|---|---|
| None | 7.71 |
| DD-VQA | 7.23 |
| FaceAPP-VQA | 6.91 |
| **FiFa-Instruct-1M (Ours)** | **6.31** |

FiFa-MLLM Baseline 2 further unfreezes the FaRL-ViT-B/16 parameters for fine-tuning. Task-specific query embeddings are introduced to support auxiliary supervision of Det. and Cls. tasks, enabling the Face Encoder to learn forgery understanding and mask prediction simultaneously. This model baseline yields comprehensive performance improvements over GLaMM. Notably, substantial gains of 5.4% and 9.0% in mIoU are observed for Artifact Mask Prediction in Loc. and AGE tasks, respectively.

The final FiFa-MLLM incorporates Region Mask Prediction auxiliary supervision. This addition further enhances performance across most tasks compared to Baseline 2, except text explanation performance in TOE and AGE. Notably, improvements are also observed in the global Det. and Cls. tasks. We attribute this enhancement to the model's improved ability to capture regional forgery clues with Region Mask Prediction auxiliary supervision.

**Data Ablation on FiFa-11.** We set the training data ratio between the Det. task and the other tasks to 1:1 for data ablation experiments, as shown in Table 7. The training data ratios among the other tasks remain consistent with those used in the multi-task FiFa-MLLM in Section 5.2 (detailed in Appendix B). For training, the Det. task exclusively utilizes data from the FFHQ and FaceAPP subsets. For evaluation, the FFHQ and FaceAPP subsets are used for Intra-Domain Det. testing, while other subsets are employed for Cross-Domain Det. testing. Experimental results demonstrate that: (1) Progressively introducing training samples for ToE, Loc., and AGE tasks, respectively, can enhance the model's Detection capability; (2) Progressively introducing Image-Level, Region-level, and Box-Level training samples from ToE, Loc., and AGE tasks, respectively, can enhance the model's Detection capability. (3) The experiment demonstrates that when incorporating all the ToE, Loc., and AGE tasks in our FiFa-11 (including all Image-Level, Region-Level, and Box-Level tasks), the model achieves the strongest DeepFake Detection capability. These findings prove that the task setup of FIFA-11 (fine-grained XDFA) can effectively enhance the MLLM's forgery understanding.

**Superiority of FiFa-Annotator.** As shown in Table 8, we jointly train Det. and I-TOE tasks with a 1:1 data ratio. The Det. task data is from FiFa-Instruct-1M, while the I-TOE data is from DD-VQA, FaceAPP-VQA, and FiFa-Instruct-1M, respectively. For fair comparison, we construct FaceAPP-VQA using an automated annotation pipeline similar to existing approaches (Xu et al., 2025; Qin et al., 2025), but with the same forgery sample sources as our FiFa-Instruct-1M. Experimental results demonstrate that: (1) Incorporating I-TOE data enhances the DeepFake Detection performance of DFA-aimed MLLMs, consistent with observations in existing works (Huang et al., 2024; Qin et al., 2025); (2) Under identical experimental settings, the I-TOE data generated by our FiFa-Annotator yields the most significant performance improvement for DeepFake Detection, demonstrating the superiority of our data pipeline over existing manual or automated approaches. The utilization of prior knowledge by FiFa-Annotator enables it to obtain more reliable data.

## 6  CONCLUSION

We propose the **Fake-in-Facext (FiFa)** framework, which enhances the fine-grained awareness of MLLMs for Explainable DeepFake Analysis (XDFA) by grounding responses in Face Visual Context. First, we define **FiFa-11**, introducing the Artifact-Grounding Explanation task and establishing novel XDFA tasks incorporating bounding box visual prompts to address a critical gap in existing literature. Second, we develop the **FiFa-Annotator** data annotation pipeline, which automatically generates precise forgery explanations with Regional Artifact Masks. Leveraging this annotator, we contribute high-quality training and testing resources to the XDFA field: **FiFa-Instruct-1M** and **FiFa-Bench**. Finally, we design and train **FiFa-MLLM** to comprehensively support all capabilities involved in FiFa-11. Our model demonstrates performance that surpasses the strong baseline by considerable margins on almost all tasks.

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

# Appendix

## Contents

# A    ADDITIONAL DETAILS FOR DATA ANNOTATION PIPELINE

## A.1    FACIAL IMAGE CONCEPT TREE

We have provided all the Atomic Concepts and Parent Concepts involved in the Facial Image Concept Tree in Tables 9 to 12.

## A.2    TOOLS OF FACIAL LANDMARK LOCALIZATION AND FACE PARSING

We briefly introduce the facial landmark localization and face parsing tools used in our FiFa-Annotator.

**Facial Landmark Localization.** We use MediaPipe Lugaresi et al. (2019), developed and open-sourced by Google Research, for facial landmark localization in face images. MediaPipe supports the prediction of 478 dense facial landmarks.

**Face Parsing.** We employ the DML-CSR Zheng et al. (2022a) model, trained on CelebA-Mask-HQ Lee et al. (2020), to perform face region segmentation. CelebA-Mask-HQ provides 19 classes, including all facial components and accessories such as skin, nose, eyes, eyebrows, ears, mouth, lip, hair, hat, eyeglass, earring, necklace, neck, and cloth.

## A.3    PROMPTS USED IN FIFA-ANNOTATOR

We provide the prompts used in the FiFa-Annotator. The prompts used in Step 3 are shown in Table 13; the prompts used in Steps 4 and 5 are shown in Table 14. In Section 5.3 of our paper, to demonstrate the superiority of the FiFa-Annotator, we construct FaceAPP-VQA using an automated annotation pipeline similar to existing approaches Xu et al. (2025); Huang et al. (2024); Qin et al. (2025), but with the same forgery sample sources as our FiFa-Instruct-1M. The prompt used for constructing FaceAPP-VQA is shown in Table 15. The prompts we used for generating detailed forgery explanations are referenced from the prompts in MMTD-SET Xu et al. (2025) when setting the analysis perspective.

## A.4    INTRODUCTION TO THE DFFD DATASET

The Diverse Fake Face Dataset (DFFD) Dang et al. (2020) is a face forgery detection dataset that covers diverse types of fake faces. Examples in the DFFD include multiple sources. Real images are from FFHQ Karras et al. (2019), CelebA Liu et al. (2015), and the YouTube subset of Face-Forensics++ Rössler et al. (2019). Face identity swapping and expression swapping examples are from the subsets of the FaceForensics++ dataset; attribute manipulation examples are from FaceAPP FaceAPP (2019) and StarGAN Choi et al. (2018); entire face synthesis samples are from PGGAN Karras et al. (2018) and StyleGAN Karras et al. (2021).

## A.5    ADDITIONAL STATISTICS OF FIFA-INSTRUCT-1M AND FIFA-BENCH

In Table 16, more detailed statistics of FiFa-Instruct-1M and FiFa-Bench are provided. Here, the FFHQ Karras et al. (2019) and CelebA Liu et al. (2015) datasets are used to provide real face image samples. YouTube, Deepfakes, FaceSwap, Face2Face, NeuralTextures, FaceAPP, StarGAN, PGGAN, and StyleGAN are all subsets of DFFD.

# B    ADDITIONAL DETAILS FOR EXPERIMENTS

**Primary Experiment for Multi-Task Setting.** During training, the LLM parameters remain frozen, with fine-tuning performed using LoRA (rank set to 128). Parameters of all other modules are set as learnable. The auxiliary tasks (Dec. and Cls.) employ cross-entropy loss; text generation uses autoregressive cross-entropy loss; mask prediction utilizes per-pixel binary cross-entropy loss combined with Dice loss. The weights of these four loss functions are empirically set to 0.2, 1.0, 0.5, and 2.0, respectively. Training is optimized using DeepSpeed Zero-2. The primary experiment (reported as FiFa-MLLM multi-task learning) is conducted on 4 NVIDIA H800 GPUs with a batch

size of 40 for 12,500 steps, employing a linear decay learning rate scheduler with a 100-step linear warm-up. The base learning rate is set to $3 \times 10^{-4}$. Table 17 details the data ratios and actual training data amounts. Configurations in experiments for Model Design Ablation for FiFa-MLLM remained consistent with the primary experiment.

**Single-Task Setting.** To ensure fair comparison, all single-task models are trained using sample quantities identical to their corresponding subsets in the multi-task setting. Each experiment utilizes one NVIDIA H800 GPU with a batch size of 10. The number of training steps is specified in Table 17 for different tasks, while other configurations align with the primary experiment.

**Experiments on DD-VQA.** The training data ratio for Det. and I-TOE is set to 4:1. The experiment employs two NVIDIA H800 GPUs with a batch size of 20. The actual training data amount for Det. corresponded to 12 epochs. Other configurations remained consistent with the primary experiment.

**Experiments on DFA-GPT.** The training data ratio for Det., Cls., and I-TOE is set to 1:1:1. The experiment employs four NVIDIA H800 GPUs with a batch size of 40. The actual training data amount for Det. corresponded to 1 epoch. Other configurations remained consistent with the primary experiment.

**Ablation Study for Data Ablation on FiFa-11.** The training data ratio between Det. and other tasks is set to 1:1. The data ratios among the other tasks remain consistent with those used in the primary experiment, as detailed in Table 17. The experiment employs one NVIDIA H800 GPU with a batch size of 10. The actual training data amount for Det. corresponded to 3 epochs. Other configurations remained consistent with the primary experiment.

**Ablation Study for the Superiority of FiFa-Annotator.** The data ratio between Det. and I-TOE is set to 1:1. The experiment employs one NVIDIA H800 GPU with a batch size of 10. The actual training data amount for Det. corresponded to 3 epochs. Other configurations remained consistent with the primary experiment.

## C  ADDITIONAL QUALITATIVE RESULTS

We provide qualitative illustrations for the Loc. and AGE tasks as shown in Figures 4-9.

## D  THE USE OF LARGE LANGUAGE MODELS

This article only uses LLM for error checking and sentence polishing.

Table 9: Facial Image Concept Tree (FICT). Due to page size limitations, we have divided FICT into four tables. This table presents concepts from Level 1 (L1) to Level 4 (L4). Atomic Concepts are highlighted in pink.

| L1 | L2 | L3 | L4 |
|---|---|---|---|
| whole facial image | foreground | region around head | region around hair
region around face |
| | | region around neck | neck
edges of neck |
| | | region around adornment and clothing | region around adornments
region around clothing |
| | background | left part of background
right part of background | |

Table 10: Sub-FICT under the Concept "region around hair (L4)". Atomic Concepts are highlighted in pink.

| L5 | L6 | L7 | L8 |
|---|---|---|---|
| hair | hair near face | hair near left part of face | hair near left part of forehead
hair near left temple
hair near left ear
hair near left cheek
hair near left part of jaws |
| | | hair near right part of face | hair near right part of forehead
hair near right temple
hair near right ear
hair near right cheek
hair near right part of jaws |
| | outer hair | left part of outer hair
right part of outer hair | |
| edges of hair | inner edge of hair
outer edge of hair | | |

Table 11: Sub-FICT under the Concept "region around adornment and clothing (L3)". Atomic Concepts are highlighted in pink.

| L4 | L5 | L6 |
|---|---|---|
| region around adornments | hat
edges of hat
eyeglasses
edges of eyeglasses | |
| | earrings | left earring
right earring |
| | edges of earrings | |
| region around clothing | clothing
edges of clothing | |

Table 12: Sub-FICT under the Concept "region around face (L4)". Atomic Concepts are highlighted in pink.

| L5 | L6 | L7 | L8 |
|---|---|---|---|
| region around forehead | forehead | left part of forehead | upper left part of forehead
lower left part of forehead |
| | | right part of forehead | upper right part of forehead
lower right part of forehead |
| | edges of forehead | top edge of forehead
bottom edge of forehead | |
| | temples | left temple
right temple | |
| | edges of temples | edges of left temple
edges of right temple | |
| | | left eyebrow | headstart of left eyebrow
tail of left eyebrow
middle part of left eyebrow
headstart of right eyebrow |

| | | | |
|---|---|---|---|
| eyebrows | | | |
| region around eyebrows | | right eyebrow | tail of right eyebrow
middle part of right eyebrow |
| | edges of eyebrows | edges of left eyebrow | top edge of left eyebrow
bottom edge of left eyebrow |
| | | edges of right eyebrow | top edge of right eyebrow
bottom edge of right eyebrow |
| | region between eyebrows | | |
| region around eyes | eyes | left eye | iris of left eye
sclera of left eye |
| | | right eye | iris of right eye
sclera of right eye |
| | edges of eyes | edges of left eye | top edge of left eye
bottom edge of left eye |
| | | edges of right eye | top edge of right eye
bottom edge of right eye |
| | region between eyes | | |
| | eyelids | eyelids of left eye | upper eyelid of left eye
lower eyelid of left eye |
| | | eyelids of right eye | upper eyelid of right eye
lower eyelid of right eye |
| | eye bags | eye bag of left eye
eye bag of right eye | |
| | eye sockets | eye sockets of left eye | upper eye socket of left eye
lower eye socket of left eye |
| | | eye sockets of right eye | upper eye socket of right eye
lower eye socket of right eye |
| region around ears | ears | left ear | upper part of left ear
lower part of left ear |
| | | right ear | upper part of right ear
lower part of right ear |
| | edges of ears | edges of left ear
edges of right ear | |
| region around nose | nose | nasion
nose bridge | |
| | | nasal ala | nasal ala of left side
nasal ala of right side |
| | | nose base | |
| | edges of nose | | |
| region around cheeks | cheeks | left cheek | upper part of left cheek
lower part of left cheek |
| | | right cheek | upper part of right cheek
lower part of right cheek |
| | edges of cheeks | edges of left cheek
edges of right cheek | |
| region around mouth | mouth | lips | upper lip
lower lip |
| | | oral cavity | |
| | edges of mouth | top edge of mouth
bottom edge of mouth | |
| region around jaws | jaws | upper jaw | left part of upper jaw
right part of upper jaw |
| | | lower jaw | left part of lower jaw
right part of lower jaw |
| | edges of jaws | top edge of jaws
bottom edge of jaws | |
| | chin | left part of chin
right part of chin | |
| | edges of chin | | |
| | philtrum | | |
| | nasolabial folds | nasolabial fold of left side
nasolabial fold of right side | |
| face edge | left part of face edge | face edge near left part of forehead
face edge near left temple
face edge near left ear
face edge near left cheek
face edge near left part of jaws | |
| | right part of face edge | face edge near right part of forehead
face edge near right temple
face edge near right ear
face edge near right cheek
face edge near right part of jaws | |

Table 13: The prompt used for Step 3 of the FiFa-Annotator. "{ concept_list }" is the list of artifact-existing Atomic Concepts.

You are an AI visual assistant that helps humans analyze some images of human faces that have been tampered with by deepfake. You will receive two images, the first is the image A of the face tampered by deepfake and the second is the binary mask image B of the tampered area (a value of 1 (white) indicates the tampered area and a value of 0 (black) indicates the untampered area).

Now, your task is to describe the visible details (artifacts) of the tampered area in the image, using the binary masks provided for the tampered regions of the deepfake tampered image A and image B. In your answer, please describe the tampered image based on the highlighted area of the binary mask, but do not mention the binary mask. Always assume that you are just looking at the tampered image. Responses should identify the tampered areas and corresponding detailed descriptions of the forgery in those areas. Use the format [tampered area]: [detailed forgery description].

All the following areas MUST be discussed:
{ concept_list }
Please analyze each area in one sentence.

When providing detailed forgery descriptions, consider the visible details caused by tampering from these perspectives, but do not give an ambiguous, unclear description that is otherwise challenging:
1. Symmetrical Facial Features: Deepfake-generated faces may exhibit unnaturally perfect symmetry, lacking the subtle asymmetry typically found in real faces.
2. Blur or Distortion Around Edges: Deepfake manipulation may introduce blur or distortion around the edges of the face where the manipulation has taken place, especially if the face has been digitally overlaid onto another body.
3. Inconsistent Lighting and Shadows: Deepfake algorithms may struggle to accurately match the lighting and shadows in the original image, leading to discrepancies in lighting direction or intensity across the face.
4. Unnatural Facial Expressions: Deepfakes may produce facial expressions that appear unnatural, exaggerated, or out of sync with the rest of the image.
5. Mismatched Facial Proportions: Deepfake manipulation may result in facial proportions that are inconsistent with the person's gender or age, such as a man's face on a woman's body or vice versa.
6. Inconsistent Skin Texture and Tone: Deepfake-generated faces may exhibit unnatural skin texture or tone, such as overly smooth or pixelated skin, that differs from the surrounding areas.
7. Missing or Inconsistent Eye Reflections: Deepfake manipulation may result in missing or inconsistent reflections in the eyes, which can provide clues about the authenticity of the image.
8. Change hairstyle: Some deepfake algorithms only tamper with hair and may change hair color and hairstyle, or add long hair for boys and short hair for girls.
9. Irregularities in Makeup Application: Deepfake manipulation may introduce makeup styles that are inconsistent with the person's gender or age, or exhibit poor application quality.
10. Contextual Inconsistencies: Deepfake-generated images may contain inconsistencies in the overall context of the image, such as discrepancies in perspective, clothing, or surroundings, that suggest manipulation.
11. Unreasonable accessories: Some deepfake algorithms add glasses, earrings, hats, masks, etc. to the image, and the edges, lighting relationships, and perspective of these accessories may be incorrect.
12. If there are glasses or sunglasses in the picture, please pay special attention to whether the glasses or sunglasses have been tampered with or not, the rims, frames, temples, lenses, etc. of the glasses are very susceptible to imperfections and problems.

Table 14: The prompt used for Steps 4 and 5 of the FiFa-Annotator.

---

Summarize the descriptions of face forgery areas into a paragraph to make it as clear as possible. The face region words in the summary can be more diverse, but the semantic consistency must be guaranteed. An index number will be provided for each face forgery area. Please indicate the index number in the summary.

Here is an example:

Input:
Descriptions
left part of chin: Displays inconsistency in skin texture, appearing artificially uniform.
right part of chin: Exhibits disproportionate shadowing inconsistent with light sources.
edges of chin: Subtle distortions suggest digital manipulation at the boundaries.
nasolabial fold of left side: Appears unnaturally softened, lacking typical depth.
nasolabial fold of right side: Exhibits lack of natural shading and depth variation.
Index Numbers
left part of chin: 0
right part of chin: 1
edges of chin: 2
nasolabial fold of left side: 3
nasolabial fold of right side: 4

Summary:
The <0> left side of the chin </0> reveals an unnaturally uniform skin texture, while the <1> right chin area </1> exhibits disproportionate shadowing that contradicts expected lighting. The <2> chin's edges </2> display subtle distortions, indicating possible digital manipulation. Additionally, the <3> left nasolabial fold </3> appears overly smoothed, lacking natural depth, whereas the <4> right nasolabial region </4> fails to show proper shading and depth variation.

Input: [input]

---

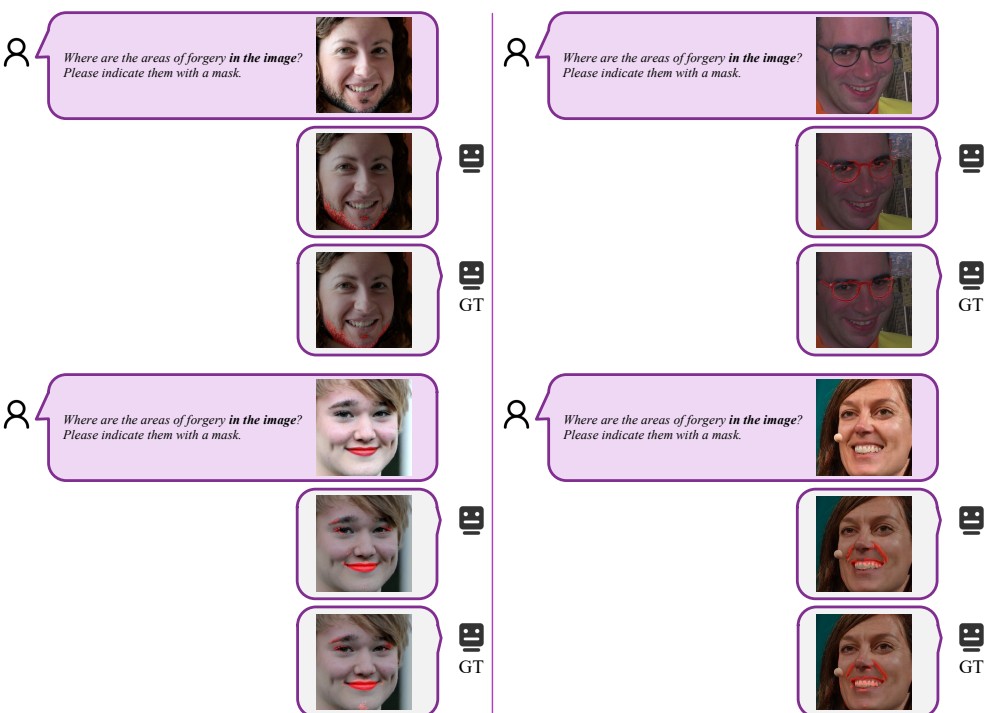

Figure 4: Output samples of our FiFa-MLLM (multi-task setting) for I-Loc.. We provide the Grounding Truth (GT) for comparison.

Table 15: The prompt used for FaceAPP-VQA.

You are an AI visual assistant that helps humans analyze some images of human faces that have been tampered with by deepfake. You will receive two images, the first is the image A of the face tampered by deepfake and the second is the binary mask image B of the tampered area (a value of 1 (white) indicates the tampered area and a value of 0 (black) indicates the untampered area).

Now, your task is to describe the visible details (artifacts) of the tampered area in the image, using the binary masks provided for the tampered regions of the deepfake tampered image A and image B. In your answer, please describe the tampered image based on the highlighted area of the binary mask, but do not mention the binary mask. Always assume that you are just looking at the tampered image. Responses should identify the tampered areas and corresponding detailed descriptions of the forgery in those areas. Use the format [tampered area]: [detailed forgery description].

Here are some phrases to describe tampered areas for reference: lower eye socket of left eye, tail of right eyebrow, eyeglasses, upper eyelid of left eye, edges of eyeglasses, region between eyes, lower eyelid of right eye, upper eye socket of right eye, top edge of left eye, lower eye socket of right eye, eye bag of right eye, upper eyelid of right eye, sclera of right eye, face edge near left temple, sclera of left eye, edges of right temple, face edge near right temple, face edge near left ear, iris of left eye, lower eyelid of left eye, nasion, upper eye socket of left eye, bottom edge of right eyebrow, edges of nose, bottom edge of right eye, bottom edge of left eye, edges of left cheek, eye bag of left eye, top edge of right eye, hair near left part of forehead, right temple, hair near left temple, inner edge of hair, face edge near left part of forehead, bottom edge of left eyebrow, face edge near right ear, face edge near left cheek, iris of right eye, top edge of right eyebrow, hair near left ear, hair near left part of jaws, edges of right cheek, upper part of left cheek, hair near left cheek, nose bridge, upper lip, oral cavity, edges of neck, upper part of right cheek, nose base, nasal ala of right side.

When providing detailed forgery descriptions, consider the visible details caused by tampering from these perspectives, but do not give an ambiguous, unclear description that is otherwise challenging:
1. Symmetrical Facial Features: Deepfake-generated faces may exhibit unnaturally perfect symmetry, lacking the subtle asymmetry typically found in real faces.
2. Blur or Distortion Around Edges: Deepfake manipulation may introduce blur or distortion around the edges of the face where the manipulation has taken place, especially if the face has been digitally overlaid onto another body.
3. Inconsistent Lighting and Shadows: Deepfake algorithms may struggle to accurately match the lighting and shadows in the original image, leading to discrepancies in lighting direction or intensity across the face.
4. Unnatural Facial Expressions: Deepfakes may produce facial expressions that appear unnatural, exaggerated, or out of sync with the rest of the image.
5. Mismatched Facial Proportions: Deepfake manipulation may result in facial proportions that are inconsistent with the person's gender or age, such as a man's face on a woman's body or vice versa.
6. Inconsistent Skin Texture and Tone: Deepfake-generated faces may exhibit unnatural skin texture or tone, such as overly smooth or pixelated skin, that differs from the surrounding areas.
7. Missing or Inconsistent Eye Reflections: Deepfake manipulation may result in missing or inconsistent reflections in the eyes, which can provide clues about the authenticity of the image.
8. Change hairstyle: Some deepfake algorithms only tamper with hair and may change hair color and hairstyle, or add long hair for boys and short hair for girls.
9. Irregularities in Makeup Application: Deepfake manipulation may introduce makeup styles that are inconsistent with the person's gender or age, or exhibit poor application quality.
10. Contextual Inconsistencies: Deepfake-generated images may contain inconsistencies in the overall context of the image, such as discrepancies in perspective, clothing, or surroundings, that suggest manipulation.
11. Unreasonable accessories: Some deepfake algorithms add glasses, earrings, hats, masks, etc. to the image, and the edges, lighting relationships, and perspective of these accessories may be incorrect.
12. If there are glasses or sunglasses in the picture, please pay special attention to whether the glasses or sunglasses have been tampered with or not, the rims, frames, temples, lenses, etc. of the glasses are very susceptible to imperfections and problems.

Table 16: Additional statistics of FiFa-Instruct-1M and FiFa-Bench.

| Task | Type | Source | Training | Dev. | Test |
|---|---|---|---|---|---|
| Det./Cls. | Live | FFHQ | 30000 | 3000 | 3000 |
| | | CelebA | 30000 | 3000 | 3000 |
| | | Youtube | 20000 | 2000 | 2000 |
| | Identity Swapping | Deepfakes | 10000 | 1000 | 1000 |
| | | FaceSwap | 10000 | 1000 | 1000 |
| | Expression Swapping | Face2Face | 10000 | 1000 | 1000 |
| | | NeuralTextures | 10000 | 1000 | 1000 |
| | Attribute Manipulation | FaceAPP | 10000 | 439 | 1000 |
| | | StarGAN | 10000 | 1000 | 1000 |
| | Entire Face Synthesis | PGGAN | 10000 | 1000 | 1000 |
| | | StyleGAN | 10000 | 1000 | 1000 |
| I-Loc. | | FaceAPP | 10000 | 439 | 1000 |
| R-Loc. | | | 300791 | 1000 | 1000 |
| B-Loc. | | | 182170 | 1000 | 1000 |
| I-TOE | | | 9940 | 438 | 992 |
| R-TOE | | | 300744 | 1000 | 1000 |
| B-TOE | | | 182097 | 1000 | 1000 |
| I-AGE | | | 9796 | 431 | 976 |
| R-AGE | | | 124113 | 1000 | 1000 |
| B-AGE | | | 103561 | 1000 | 1000 |
| Total | | | 1383212 | 22747 | 24968 |

Table 17: The data ratios and actual training data amounts in the multi-task setting. The training steps for different tasks in the single-task setting.

| Task | Data Ratio | Data Amount in FiFa-Instruct-1M | Training Data Amount for Multi-Task Model | Actual Epoch | Training Steps for Single-Task Models |
|---|---|---|---|---|---|
| Det. | 4 | 160000 | 55556 | 0.35 | 5555 |
| Cls. | 4 | 160000 | 55556 | 0.35 | 5555 |
| I-TOE | 1 | 9940 | 13888 | 1.4 | 1388 |
| I-Loc. | 1 | 10000 | 13888 | 1.39 | 1388 |
| I-AGE | 2 | 9796 | 27776 | 2.84 | 2777 |
| R-TOE | 3 | 300744 | 41668 | 0.14 | 4166 |
| R-Loc. | 3 | 300791 | 41668 | 0.14 | 4166 |
| R-AGE | 6 | 124113 | 83332 | 0.67 | 8333 |
| B-TOE | 3 | 182097 | 41668 | 0.23 | 4166 |
| B-Loc. | 3 | 182170 | 41668 | 0.23 | 4166 |
| B-AGE | 6 | 103561 | 83332 | 0.8 | 8333 |

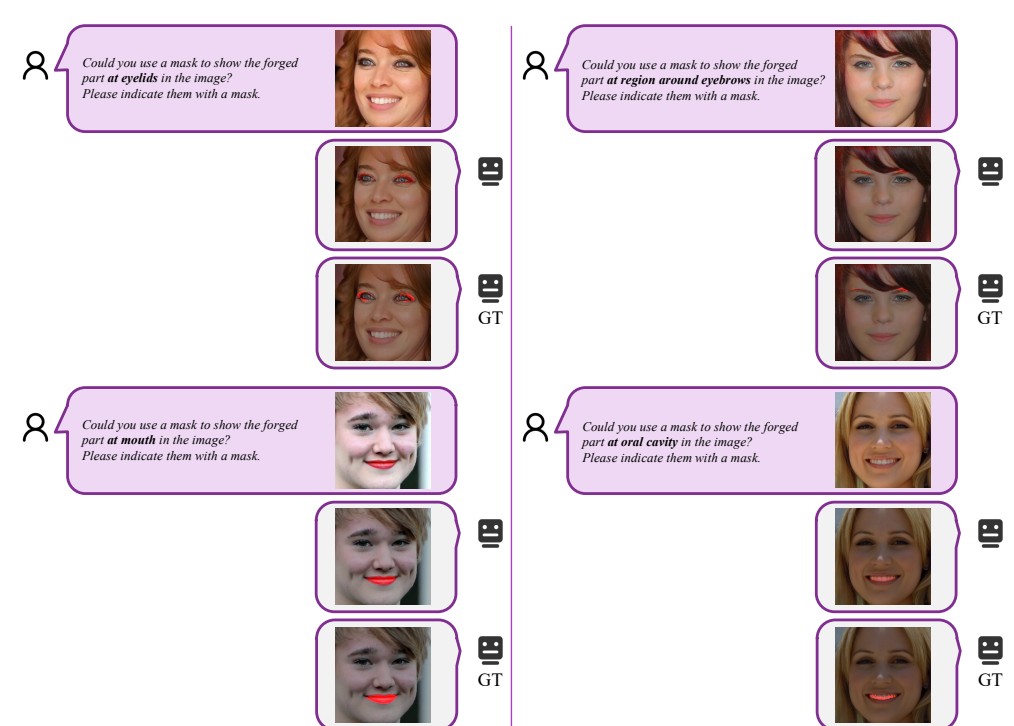

Figure 5: Output samples of our FiFa-MLLM (multi-task setting) for R-Loc.. We provide the Grounding Truth (GT) for comparison.

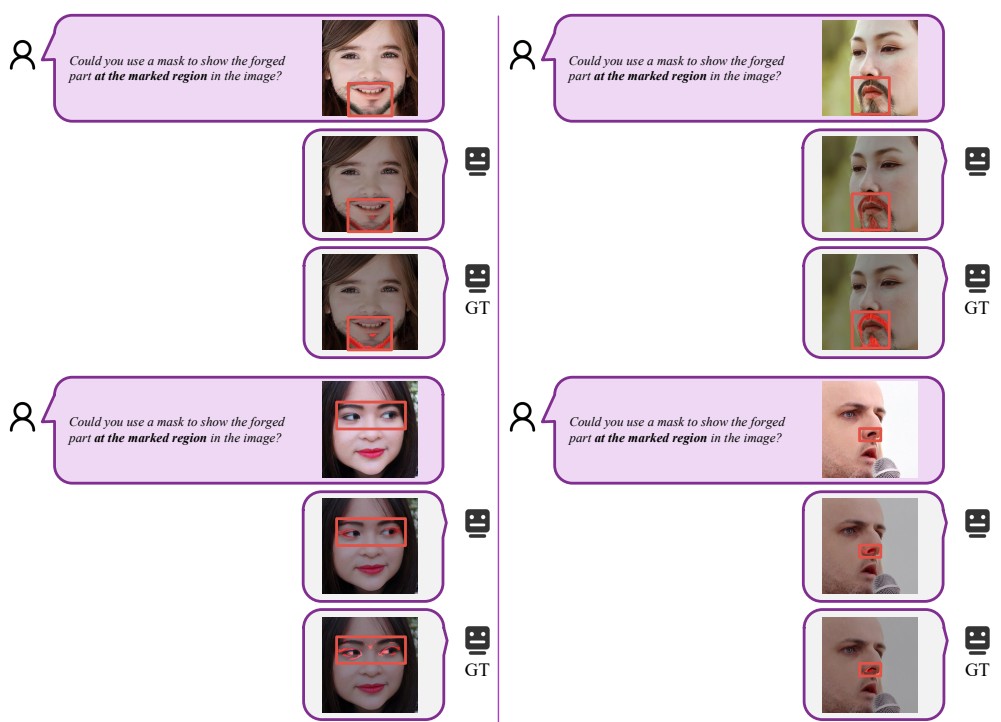

Figure 6: Output samples of our FiFa-MLLM (multi-task setting) for B-Loc.. We provide the Grounding Truth (GT) for comparison.

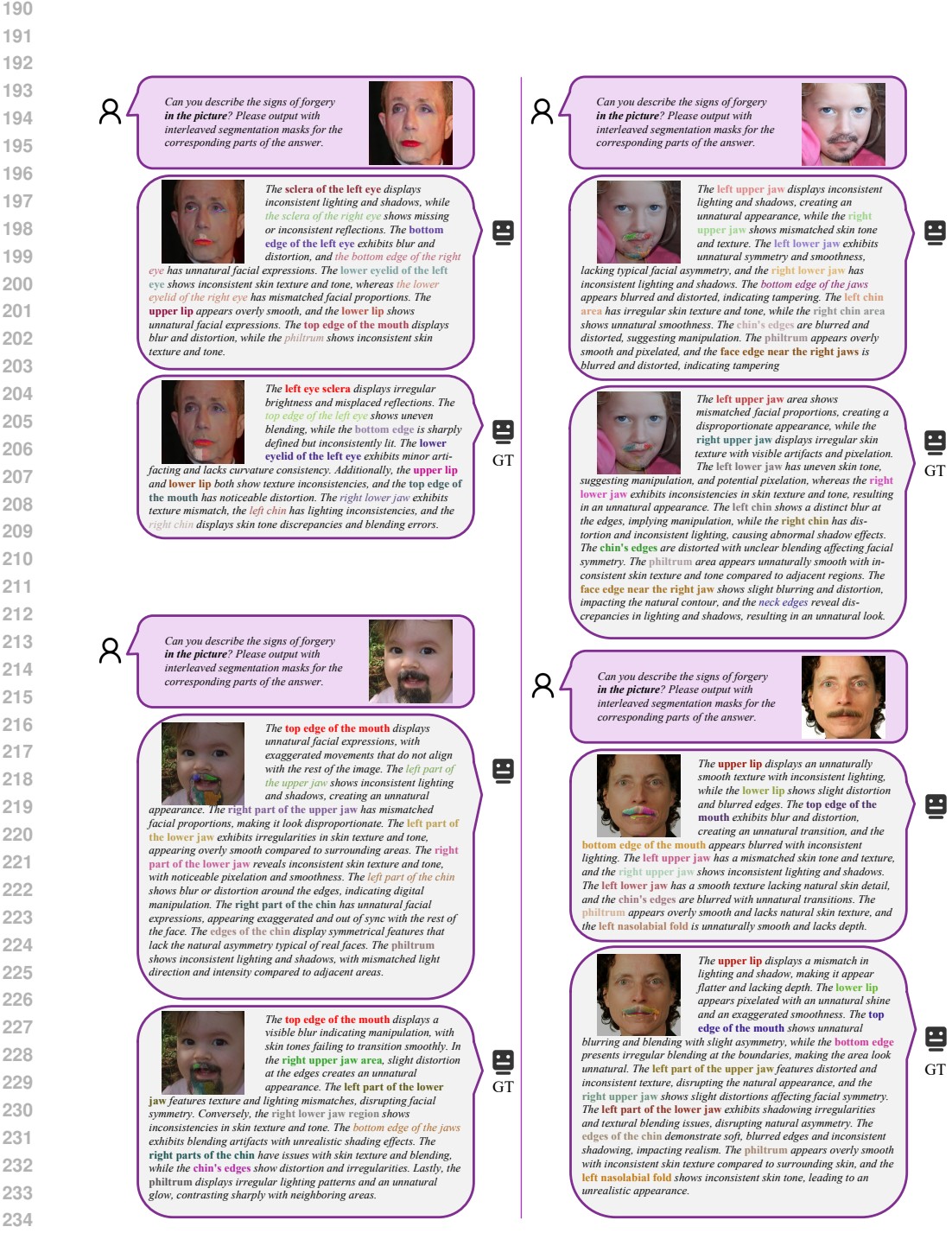

Figure 7: Output samples of our FiFa-MLLM (multi-task setting) for I-AGE. We provide the Grounding Truth (GT) for comparison. In the textual explanations, the recalled Atomic Concepts are highlighted in **bold**.

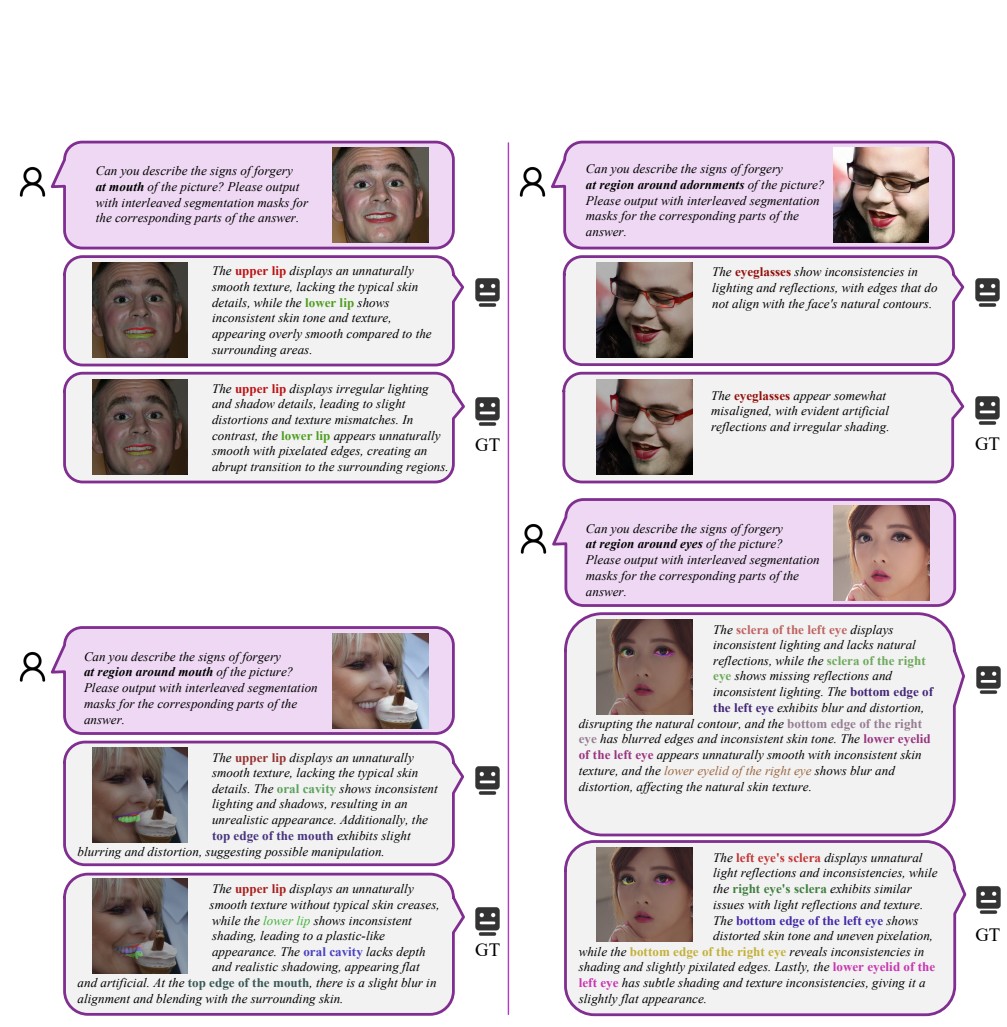

Figure 8: Output samples of our FiFa-MLLM (multi-task setting) for R-AGE. We provide the Grounding Truth (GT) for comparison. In the textual explanations, the recalled Atomic Concepts are highlighted in **bold**.

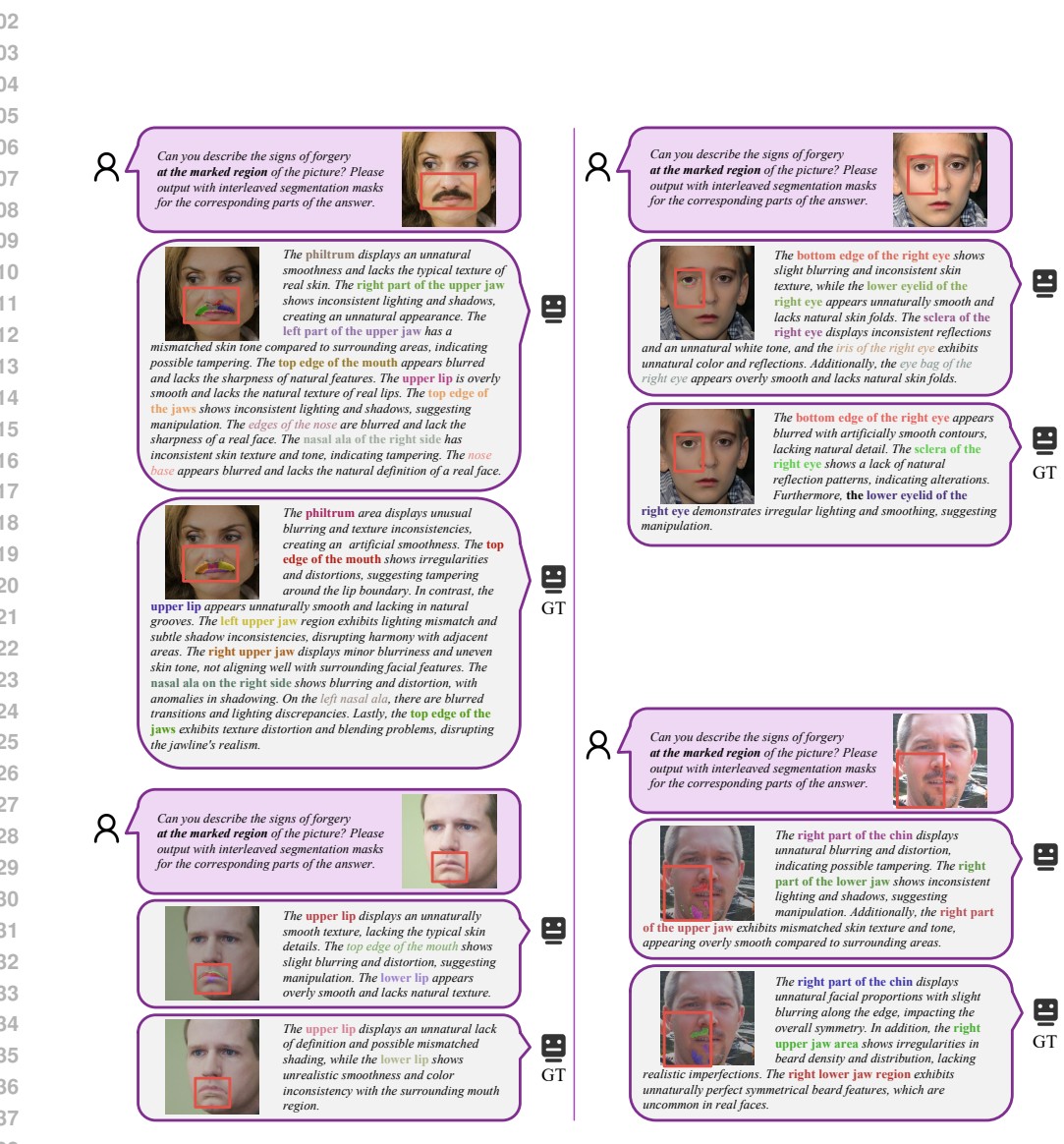

Figure 9: Output samples of our FiFa-MLLM (multi-task setting) for B-AGE. We provide the Grounding Truth (GT) for comparison. In the textual explanations, the recalled Atomic Concepts are highlighted in **bold**.

