# OpenReview forum: "Fake-in-Facext: Towards Fine-Grained Explainable DeepFake Analysis"
_ICLR.cc/2026/Conference — Submitted to ICLR 2026_

### Official Review · Reviewer_Vjsn · 2025-10-24

**Soundness:** 3
**Presentation:** 4
**Contribution:** 4
**Rating:** 6
**Confidence:** 4

**Summary:**

The paper proposes an MLLM with multiple input streams and multiple outputs (natural language, RGB and segmentation/bounding box) for deepfake detection, localisation and explanability. To achieve this, the paper first proposes an annotation method and creates a 1+M image dataset, with fine-grained annotations and hierarchical labels. Then, to standardise and ensure safety, a benchmarking for the new task is proposed. Finally, an MLLM is trained on the task using the proposed dataset and evaluated against the baseline, and evaluated on the proposed benchmark and DDVQA.

**Strengths:**

The paper is well written and easy to follow with little, if any, ambiguity. The problem identified is largely with existing datasets (i.e. they largely have binary labels), and the authors fix it by proposing a new annotation method, constructing a very large-scale dataset and showing that MLLMs can adequately learn and perform in the downstream tasks of deepfake detection and localisation. Contrary to previous works (eg, DDVQA, which is the main comparison in the paper), the MLLM has multiple outputs to enhance natural language explanations, which is both novel in the task using MLLMs and intuitive (natural language and visual explanations are complementary).

**Weaknesses:**

The main weakness identified is the lack of experiments on previously established and standard deepfake detection datasets (eg, FF++, DFDC, Wild DeepFake, etc), and subsequent comparison with non-MLLM methods on these. This is essential both for context and to assess the generalisation capabilities of the proposed MLLM, particularly as the explanation in sec. 2.2, step 1 implies that the forgery methods used are rather limited.

Some minor comments:
- Table 4 is unclear whether the MLLM is trained from scratch on DDVQA, fine-tuned or if this is cross-dataset performance.
- In the Multi-task processing, the sections describing samples are confusing: "For the sample Det., T_in is the detection embedding [...]" -> this and the subsequent sentences are not making a lot of sense; it would be best to rephrase for clarity.
- In method 2.2, step 2, again, the explanation for atomic/parent concepts is a little unclear. Consider maybe breaking down and expanding on the conditions for clarity.
- Some Qualitative examples and Human evaluation results on the natural language output would strengthen the paper

**Questions:**

- What is the cross-dataset and zero-shot performance of the model on standard DF detection datasets?
- Are the Table 4 results cross-dataset, finetuned or trained from scratch?
- How do humans rate the natural language output of the MLLM?

---

> ### Author Response · Authors · 2025-11-16
> **For Major Weakness and Question 1**
>
> First, we explain why we did not report the performance of our FiFa-MLLM on previously established deepfake detection datasets.
> Our work is the first to explore the Artifact-Grounding Explanation (AGE) task, aiming to ground the concept of Artifact-Existing regions in forgery explanations to segmentation masks. We pursue the interpretability of forgery traces at the human perception level, requiring forgery localization consistent with human common sense. Specifically, it involves identifying the most unnatural regions in images, describing them in text, and providing corresponding mask annotations. This heightened pursuit of interpretability means we need to balance the performance of forgery Detection, Classification, Localization, Text-Only Explanation, and Artifact-Grounding Explanation during training, rather than deliberately optimizing the performance of forgery detection as a binary classification task.
> Thus, it would be unfair to directly compare our method with existing approaches on previously established binary classification datasets (e.g., FF++, DFDC, and Wild DeepFake). However, on DD-VQA and DFA-Bench, constructed for Explainable DeepFake Analysis, we have achieved SOTA results on the forgery detection task, demonstrating the potential of our model.
> In the future, we will try to obtain more computational resources to explore strategies such as extending training steps or adding other supervision losses to pursue higher performance in both forgery detection and interpretability metrics simultaneously.

---

> > ### Comment · Reviewer_Vjsn · 2025-11-18
> >
> > Thank you for the very detailed response and for taking the time to respond to each comment, this is appreciated. I will try to discuss each point in a single comment to keep the discussion coherent.
> >
> > - Cross-dataset generalisation: I understand the point from an annotation point of view, however, at the end of the day DD-VQA is a subset of FF++ with natural language annotations. As such, there should not be a significant domain shift in terms of the input. Furthermore, the detection task within dataset achieves near perfect accuracy even for traditional methods, making it more or less trivial. The real performance of any DF method can only be seen cross-dataset and in-the-wild. As such, I maintain the original point, that such experiments and comparisons are not only feasible but also necessary.
> >
> > - Localisation: This is understandable given the field, but again ties to the previous point regarding generalisation: where does the method fail? works like SBI even, do indeed manipulate attributes of essentially face-swap, however, this is done to achieve better results on the binary task that also generalise to unseen samples.
> >
> > Overall, I still think this is not a bad paper, but as DF is saturated as a task with even simple binary methods achieving very high accuracy, some comparison out of domain and essentially placing this work in the wider context seems necessary.

---

> ### Author Response · Authors · 2025-11-16
> **For Major Weakness and Question 1 (continued from above)**
>
> Second, we elaborate on the limitations regarding the use of forgery techniques. For the forgery detection task, we have covered four types of forgery techniques: identity swapping, expression swapping, attribute manipulation, and entire face synthesis samples. This point has been illustrated in Section 3.3 of the main text and Appendix A.4. Regarding the incorporation of multiple forgery techniques in the forgery mask prediction, we believe the existing literature lacks sufficient referable solutions. We have made the most active attempts within the scope of existing technologies.
> The specific analysis is as follows:
> Currently, existing forgery localization techniques adopt two methods for constructing forgery masks:
> 1. Treating all pixels or boundary pixels of the entire forged facial region as the forgery mask, as in References [1]-[4]. The motivation behind these methods is that some early face forgery techniques cut out the facial region to be forged and replace it with a newly generated one. These forgery methods are "whole-face level" and clearly cannot meet the needs of fine-grained face forgery localization in our work.
> 2. Calculating pixel intensity differences between real images and forged images followed by binarization to obtain the forgery mask, as in References [5]-[6]. This method is also adopted in our work.
>
> It should be noted that existing non-MLLM-based forgery localization works mainly serve as auxiliary tasks for forgery detection to improve the performance of binary classification in forgery detection. Therefore, they do not care whether the localized forged regions are consistent with human perception.
>
> Our work is the first to explore the Artifact-Grounding Explanation (AGE) task, aiming to ground the concept of Artifact-Existing regions in forgery explanations to segmentation masks. We pursue the interpretability of forgery traces at the human perception level, requiring forgery localization consistent with human common sense.
>
> From an empirical perspective, attribute manipulation concentrates unnatural traces in the altered regions. You can refer to the image samples presented in the main text and the appendix. It can be observed that in the FaceAPP data we used, the regions with the largest differences in the forgery masks are visually consistent with human common sense of forgery.
>
> However, this is not the case for forged images generated through identity swapping or expression swapping. Due to significant changes in identity and expression, the positions of facial concepts in forged images shift significantly from those in original images. Consider the following scenario: in identity swapping, the eyes of the source image are relatively above those of the target image, leading to the eyes in the forged image also being above those in the original real image. Calculating pixel differences at this time will leave traces of two pairs of eyes (because they are misaligned) in the forgery mask. Nevertheless, we cannot confirm that these regions with the largest differences in the forgery mask are necessarily the most visually unnatural. A similar issue exists in expression swapping.
>
> We acknowledge that relying solely on attribute manipulation as the source of data for forgery localization and the AGE task limits the capabilities of our model. However, based on our investigation, there is currently no more effective method to obtain forgery localization masks consistent with human common sense automatically. As an exploratory work, we aim to prioritize ensuring data quality rather than hastily attempting to cover all four types of forgery techniques in the absence of a mature automated solution. Therefore, we only use data from attribute manipulation to construct artifact masks. In fact, another recent work, FAKESHIELD [6] (which supports Image-Level Localization), also only supports forgery localization for the attribute manipulation task and abandons other forgery types.

---

> ### Author Response · Authors · 2025-11-16
> **References**
>
> [1] Nguyen, H. H., Fang, F., Yamagishi, J., & Echizen, I. (2019, September). Multi-task learning for detecting and segmenting manipulated facial images and videos. In 2019 IEEE 10th international conference on biometrics theory, applications and systems (BTAS) (pp. 1-8). IEEE.
>
> [2] Kong, C., Chen, B., Li, H., Wang, S., Rocha, A., & Kwong, S. (2022). Detect and locate: Exposing face manipulation by semantic-and noise-level telltales. IEEE Transactions on Information Forensics and Security, 17, 1741-1756.
>
> [3] Lai, Y., Luo, Z., & Yu, Z. (2023, December). Detect any deepfakes: Segment anything meets face forgery detection and localization. In Chinese conference on biometric recognition (pp. 180-190). Singapore: Springer Nature Singapore.
>
> [4] Li, L., Bao, J., Zhang, T., Yang, H., Chen, D., Wen, F., & Guo, B. (2020). Face x-ray for more general face forgery detection. In Proceedings of the IEEE/CVF conference on computer vision and pattern recognition (pp. 5001-5010).
>
> [5] Dang H, Liu F, Stehouwer J, et al. On the detection of digital face manipulation[C]//Proceedings of the IEEE/CVF Conference on Computer Vision and Pattern recognition. 2020: 5781-5790.
>
> [6] Xu, Z., Zhang, X., Li, R., Tang, Z., Huang, Q., & Zhang, J. FakeShield: Explainable Image Forgery Detection and Localization via Multi-modal Large Language Models. In The Thirteenth International Conference on Learning Representations.

---

> ### Author Response · Authors · 2025-11-16
> **For Minor Weakness and Other Questions**
>
> ### For Minor Weakness 1 and Question 2
> When evaluated on the DD-VQA dataset, FiFa-MLLM is trained from scratch using only the training data of DD-VQA. The parameter initialization method is consistent with that in the main experiment, which can be referred to in Section 5.1. We will clarify this detail in the revised manuscript.
>
> ### For Minor Weakness 2
> What we intend to convey is that $ T_{\text{in}} $ is a variable input. During training, different embeddings are selected based on the current training sample:
> - When the current training sample is a Detection (Det.) task sample, $ T_{\text{in}} $ is set to the detection embedding $D \in \mathbb{R}^{1 \times D_{\text{de}}} $;
> - When it is a Classification (Cls.) task sample, $ T_{\text{in}} $ is set to the classification embedding $ C \in \mathbb{R}^{1 \times D_{\text{de}}} $;
> - When it is a Localization (Loc.) or Artifact-Grounding Explanation (AGE) task sample, $ T_{\text{in}} $ is set to the mask embedding $M \in \mathbb{R}^{1 \times D_{\text{de}}} $ and the semantic embedding $ S \in \mathbb{R}^{1 \times D_{\text{de}}} $.
>
> We will revise the expression of this section in the revised manuscript to facilitate readers' understanding.
>
> ### For Minor Weakness 3
> Atomic/Parent Concepts are the concepts introduced in Section 3.1. To model context in facial images, we propose the Facial Image Concept Tree (FICT), depicted in Figure 2(a). This hierarchical tree comprises 8 levels, where each node represents a region concept within the facial image. We designate concepts on leaf nodes as Atomic Concepts and concepts on internal nodes as Parent Concepts.
>
> ### For Minor Weakness 4 and Question 3
> We have provided numerous qualitative examples in Appendix C to facilitate the understanding of our model’s capabilities. These visualization examples provide accurate artifact mask predictions and text explanations, and align both of them correctly.

---

> ### Author Response · Authors · 2025-11-29
>
> Thank you for your valuable suggestions. We would like to provide additional clarifications on the cross-dataset generalization experiments.
> We acknowledge your observation that "the detection task within dataset achieves near perfect accuracy even for traditional methods". However, traditional methods typically do not incorporate large-scale parameter MLLMs, and more importantly, they are not required to simultaneously support 11 fine-grained forgery analysis tasks with distinct inputs and outputs. Therefore, it would be unfair to require us to compare our performance directly with these traditional methods.
>
> Since the core objective of our paper is to establish connections between textual forgery explanations and the visual evidence of artifacts, as well as to support queries for arbitrary facial regions, we prioritized demonstrating the effectiveness of our proposed model through the main experiments presented in Table 3.
> Given the limited training computational resources, we aimed to ensure decent performance across all 11 supported tasks, which led us to set the task sample ratios and task weights as detailed in Appendix B.
>
> The massive parameters of MLLMs should be leveraged to support complex explainable tasks, which traditional architectures are inherently unable to handle, rather than focusing solely on comparing forgery performance with specialized models under the traditional paradigm.
>
> In fact, there are some plug-and-play methods for independently improving forgery detection performance, such as incorporating an additional forged face image encoder and directly utilizing its features for the binary classification output head. Examples include CLIP-ViT integrated into the MIDS module in FFAA [1] and the DeepFake Encoder adopted in M2F2-Det [2].  However, they are not the focus of our study. We may explore such plug-and-play approaches in our future work.
>
> Furthermore, our experiments have shown that increasing the number of training steps significantly improves the model's performance across all tasks. In the future, we will try to obtain more computational resources to verify the upper bound of facial forgery detection performance when all tasks are trained jointly.
>
> Regarding the generalization for the Localization task, as elaborated in our response to Major Weaknesses, there is currently a lack of appropriate techniques to extract forgery localization masks aligned with human common sense for images generated by methods such as identity swapping, expression swapping, or entire face synthesis. Consequently, we are now unable to evaluate the model's performance in these scenarios.
>
> [1] Huang, Z., Xia, B., Lin, Z., Mou, Z., Yang, W., & Jia, J. (2024). Ffaa: Multimodal large language model based explainable open-world face forgery analysis assistant. arXiv preprint arXiv:2408.10072.
>
> [2] Guo, X., Song, X., Zhang, Y., Liu, X., & Liu, X. (2025). Rethinking Vision-Language Model in Face Forensics: Multi-Modal Interpretable Forged Face Detector. In Proceedings of the Computer Vision and Pattern Recognition Conference (pp. 105-116).

---

### Official Review · Reviewer_nV3V · 2025-10-24

**Soundness:** 3
**Presentation:** 3
**Contribution:** 3
**Rating:** 6
**Confidence:** 4

**Summary:**

This paper presents Fake-in-Facext (FiFa), a comprehensive framework for fine-grained Explainable DeepFake Analysis (XDFA). The authors introduce a Facial Image Concept Tree (FICT) to model facial regions for fine-grained artifact localization, an automated annotation pipeline to generate image-text pairs with explanations, and a unified multi-task model FiFa-MLLM for joint predictions of textual forgery explanations and segmentation masks. Extensive experiments show that FiFa-MLLM achieves state-of-the-art performance in artifact grounding and localization.

**Strengths:**

1. The work makes progress toward fine-grained explainability in deepfake analysis through textual and visual artifact reasoning.

2. The hierarchical concept tree and annotation pipeline are well-motivated and improve annotation precision.

3. The unified architecture for multimodal framework is well-designed, which integrates multi-task learning without requiring multiple encoders.

4. Experimental results across multiple datasets demonstrates the framework’s effectiveness and data reliability.

**Weaknesses:**

Major Weaknesses

1.	The semantic consistency between generated textual explanations and segmentation masks is not deeply validated. While the multi-task decoders output both modalities, the linguistic and visual results may be unaligned.

2.	The FiFa-Annotator pipeline heavily relies on GPT-4o and ChatGPT for generating explanations, which may introduce linguistic or conceptual bias. The authors claim reliability improvement via prior knowledge but do not quantify annotation quality beyond model performance.

3.	The qualitative results for Artifact-Grounding Explanation (AGE) are not extensively illustrated. The paper would benefit from more visual samples showing how textual and segmentation outputs correspond at different facial content.

Minor Weaknesses

1.	Some concept names (e.g., “Facext”, “Artifact-Grounding Explanation”) can be clearly defined for better readability.

2.	The introduction section could clarify the novelty of FiFa-MLLM relative to prior segmentation-aware MLLMs.

**Questions:**

1. How do the authors verify that textual explanations and segmentation masks are semantically aligned?

2. How is the annotation quality from GPT-4o and ChatGPT controlled or measured in FiFa-Annotator?

3. Could the authors add more visual examples to show text–mask correspondence in the AGE results?

4. How does FiFa-MLLM differ architecturally from prior segmentation-aware MLLMs such as LISA[1] or GLaMM[2]?

[1] Xin Lai, Zhuotao Tian, Yukang Chen, Yanwei Li, Yuhui Yuan, Shu Liu, and Jiaya Jia. LISA: reasoning segmentation via large language model. In CVPR, pp. 9579–9589. IEEE, 2024.

[2] Hanoona Abdul Rasheed, Muhammad Maaz, Sahal Shaji Mullappilly, Abdelrahman M. Shaker, Salman H. Khan, Hisham Cholakkal, Rao Muhammad Anwer, Eric P. Xing, Ming-Hsuan Yang, and Fahad Shahbaz Khan. Glamm: Pixel grounding large multimodal model. In CVPR, pp. 13009–13018. IEEE, 2024.

---

> ### Author Response · Authors · 2025-11-16
>
> Thank you for your insightful questions.
>
> ### For Major Weakness 1 and Question 1
> First, in terms of the model architecture, each of our segmentation mask predictions is derived from decoding the special token [SEG] following each regional concept in the text, which structurally ensures the semantic consistency between text explanations and segmentation masks.
>
> Second, regarding qualitative validation, we have provided visualizations of AGE task samples in Appendix C (as shown in Figures 7-9. It can be observed that text explanations are aligned with segmentation masks.
>
> Finally, for quantitative validation, our test set (i.e., the ground truth for mask prediction) strictly guarantees the semantic consistency between text explanations and segmentation masks. In fact, the calculation of mIoU can quantitatively reflect the semantic consistency between the generated text explanations and segmentation masks. As our work is the first to introduce the AGE task, we compared performance with the strong baseline GLaMM, demonstrating that our proposed FiFa-MLLM can achieve superior performance over GLaMM on the mask prediction subtask of AGE.
>
> ### For Major Weakness 2 and Question 2
> We directly identify regions with forgeries by introducing prior knowledge (i.e., attribute manipulation concentrates unnatural traces in modified regions) instead of leaving this determination to GPT-4o, thereby ensuring the reliability of the forgery descriptions generated by GPT-4o.
>
> Specifically, existing works [1-3] generally input original images and forged images directly, requiring GPT-4o to independently discover and analyze forgery traces to construct training data. In contrast, our method does not require GPT-4o to autonomously detect regions with forgery traces. Instead, we first compute the forgery masks, determine the Artifact-Existing Concepts following Step 2 of Section 3.2, and use the list of Artifact-Existing Concepts as a constraint to guide GPT-4o to focus on these regions—this approach enhances the accuracy of the forgery descriptions generated by GPT-4o.
>
> Next, we will measure the "annotation quality" in our annotation pipeline.
> As noted in Section 5 of the paper, for fair comparison, we constructed FaceAPP-VQA using an automated annotation pipeline similar to existing approaches, and we also provided the prompt used to construct FaceAPP-VQA in Table 15. This prompt instructs GPT-4 to determine regions containing forgeries. We randomly extracted 100 samples with identical image IDs from both FaceAPP-VQA and the I-AGE task subset of FiFa-Instruct-1M and manually inspected whether the regional concepts included in their textual explanations are consistent with the artifact masks.
>
> Upon inspection, the 100 samples from FiFa-Instruct-1M contained a total of 889 regional concepts, with artifact masks present for all corresponding regions. In contrast, the 100 samples from FaceAPP-VQA contained 612 regional concepts, among which only 383 had actual artifact masks—accounting for 62.6% of the total. This confirms that our annotation pipeline indeed exhibits better annotation quality.
>
> ### For Major Weakness 3 and Question 3
> We present numerous qualitative results of the Artifact-Grounding Explanation (AGE) task in Figures 7–9 of Appendix C.
>
> ### For Minor Weakness 1
> The term Facext refers to Face Visual Context, and we will denote the origin of the abbreviation with an underscore in the revised version.
>
> Artifact-Grounding Explanation refers to the task of generating textual forgery explanations interleaved with segmentation masks of manipulated artifacts. This task has been defined in the abstract and introduction, and illustrated with visual examples in Figure 1.
>
> ### For Minor Weakness 2 and Question 4
> Compared with previous segmentation-capable MLLMs (e.g., LISA and GLaMM), our FiFA-MLLM employs only one global visual encoder. This encoder concurrently generates appropriate visual features for both LLM input and mask prediction. Furthermore, we propose a Multi-Task Decoder. By introducing distinct task-specific query embeddings, this decoder can simultaneously handle Artifact Mask Prediction and multiple auxiliary supervision tasks. We have clarified our structural innovations in the introduction.
>
> ### References
> [1] Xu, Z., Zhang, X., Li, R., Tang, Z., Huang, Q., & Zhang, J. FakeShield: Explainable Image Forgery Detection and Localization via Multi-modal Large Language Models. In The Thirteenth International Conference on Learning Representations.
>
> [2] Huang, Z., Xia, B., Lin, Z., Mou, Z., Yang, W., & Jia, J. (2024). Ffaa: Multimodal large language model based explainable open-world face forgery analysis assistant. arXiv preprint arXiv:2408.10072.
>
> [3] Qin, L., Jiang, N., Zhang, Y., Qiu, Y., Zeng, D., Hu, J., & Deng, W. (2025, April). Towards Interactive Deepfake Analysis. In ICASSP 2025-2025 IEEE International Conference on Acoustics, Speech and Signal Processing (ICASSP) (pp. 1-5). IEEE.

---

### Official Review · Reviewer_iCL8 · 2025-10-31

**Soundness:** 3
**Presentation:** 2
**Contribution:** 3
**Rating:** 6
**Confidence:** 5

**Summary:**

This paper proposes the Fake-in-Facext (FiFa) framework to address the lack of fine-grained awareness in existing multimodal large language models for explainable deepfake analysis. The framework introduces a hierarchical Facial Image Concept Tree with 112 atomic concepts for reliable data annotation, defines 11 tasks (FiFa-11) including a novel Artifact-Grounding Explanation task that interleaves textual explanations with segmentation masks, and constructs FiFa-Instruct-1M, the largest training dataset (1.38M samples) in the XDFA field.

**Strengths:**

1. The paper leverages multimodal large language models for deepfake annotation and considers diverse tasks including bounding box-level queries, which advances the explainability of deepfake detection. The multi-granularity task design (image-level, region-level, and box-level) enables fine-grained forgery analysis.

2. The authors contribute a datasetwith comprehensive annotations covering 11 different tasks, including novel artifact-grounding explanations that interleave textual descriptions with segmentation masks. This represents the largest training dataset in the explainable deepfake analysis field to date (1.38M QA-pairs).

3. The paper proposes a reasonable baseline model (FiFa-MLLM) for comprehensive evaluation. The unified architecture with a single visual encoder for both LLM input and mask prediction demonstrates efficiency, and the ablation studies validate the effectiveness of key components such as auxiliary supervision tasks.

**Weaknesses:**

1. The paper only considers attribute manipulation techniques (primarily FaceApp) for creating fake samples, excluding other common deepfake types such as identity swapping, expression swapping, and entire face synthesis in the data annotation pipeline (FiFa-Annotator). This narrow focus on a single forgery method may hinder the model's generalization capability to diverse deepfake techniques encountered in real-world scenarios.

2. The data generation approach (using masks + large language models) appears similar to the CVPR 2025 paper "Towards General Visual-Linguistic Face Forgery Detection", which also analyzes fine-grained facial features and uses them for large model context learning. While the proposed Facial Image Concept Tree provides more detailed annotations (112 atomic concepts vs. coarser regions), it remains unclear whether this increased granularity translates to meaningful performance gains. I recommend the authors conduct a direct comparison with this method, given the similar annotation paradigms, to demonstrate the added value of the fine-grained concept hierarchy.

3. The evaluation is relatively limited and narrow in scope. The experiments primarily focus on performance on the authors' own dataset (FiFa-Bench) with limited baseline comparisons (mainly GLaMM). Critically, there is insufficient evaluation of generalization performance on out-of-distribution datasets or real-world deployment scenarios. Given the narrow data range (only FaceApp for training AGE/TOE/Loc tasks), I have significant concerns about whether the proposed method can generalize to other deepfake generation techniques, diverse manipulation methods, or in-the-wild forgery detection. Additional cross-dataset evaluation (e.g., on FaceForensics++, Celeb-DF, or DFDC) would strengthen the paper's claims.

4. The paper claims SOTA results on DD-VQA and DFA-Bench (Tables 4-5), but these improvements are relatively modest. More diverse baseline comparisons with recent XDFA methods (FFAA, FakeShield, etc.) would better contextualize the contributions.

Despite the aforementioned issues, I acknowledge that the authors' proposed comprehensive multimodal evaluation framework, particularly the inclusion of bounding box-level regression tasks, makes a meaningful contribution to this field. I hope the author can address my concerns.

**Questions:**

1. Why 112 atomic concepts specifically? No ablation on the granularity of FICT (e.g., 50 vs. 112 vs. 200 concepts)

2. This paper replacing CLIP+SAM encoders with a single FaRL encoder. It's unclear whether improvements come from the unified encoder design, face-specific pretraining (FaRL), or simply better training data

3. Table 6 shows region mask prediction helps, but: Improvements are relatively modest (Loc: 30.0→30.9 mIoU, AGE: 30.0→30.4 mIoU)
Det/Cls auxiliary tasks actually hurt some metrics (TOE METEOR drops from 21.3 to 20.9 without region mask)

---

> ### Author Response · Authors · 2025-11-16
> **For Weakness 1**
>
> Thank you for your recognition of our paper and your insightful questions. We will provide a detailed response to your concerns in the following sections.
>
> ### For Weakness 1
> We will elaborate on the limitations regarding the use of forgery techniques. For the forgery detection task, we have covered four types of forgery techniques: identity swapping, expression swapping, attribute manipulation, and entire face synthesis samples. This point has been illustrated in Section 3.3 of the main text and Appendix A.4. Regarding the incorporation of multiple forgery techniques in the forgery mask prediction, we believe the existing literature lacks sufficient referable solutions. We have made the most active attempts within the scope of existing technologies. The specific analysis is as follows:
>
> Currently, existing forgery localization techniques adopt two methods for constructing forgery masks:
> 1. Treating all pixels or boundary pixels of the entire forged facial region as the forgery mask, as in References [1]-[4]. The motivation behind these methods is that some early face forgery techniques cut out the facial region to be forged and replace it with a newly generated one. These forgery methods are "whole-face level" and clearly cannot meet the needs of fine-grained face forgery localization in our work.
> 2. Calculating pixel intensity differences between real images and forged images followed by binarization to obtain the forgery mask, as in References [5]-[6]. This method is also adopted in our work.
>
> It should be noted that existing non-MLLM-based forgery localization works mainly serve as auxiliary tasks for forgery detection to improve the performance of binary classification in forgery detection. Therefore, they do not care whether the localized forged regions are consistent with human perception.
>
> Our work is the first to explore the Artifact-Grounding Explanation (AGE) task, aiming to ground the concept of Artifact-Existing regions in forgery explanations to segmentation masks. We pursue the interpretability of forgery traces at the human perception level, requiring forgery localization consistent with human common sense.
>
> From an empirical perspective, attribute manipulation concentrates unnatural traces in the altered regions. You can refer to the image samples presented in the main text and the appendix. It can be observed that in the FaceAPP data we used, the regions with the largest differences in the forgery masks are visually consistent with human common sense of forgery.
>
> However, this is not the case for forged images generated through identity swapping or expression swapping. Due to significant changes in identity and expression, the positions of facial concepts in forged images shift significantly from those in original images. Consider the following scenario: in identity swapping, the eyes of the source image are relatively above those of the target image, leading to the eyes in the forged image also being above those in the original real image. Calculating pixel differences at this time will leave traces of two pairs of eyes (because they are misaligned) in the forgery mask. Nevertheless, we cannot confirm that these regions with the largest differences in the forgery mask are necessarily the most visually unnatural. A similar issue exists in expression swapping.
>
> We acknowledge that relying solely on attribute manipulation as the source of data for forgery localization and the AGE task limits the capabilities of our model. However, based on our investigation, there is currently no more effective method to obtain forgery localization masks consistent with human common sense automatically. As an exploratory work, we aim to prioritize ensuring data quality rather than hastily attempting to cover all four types of forgery techniques in the absence of a mature automated solution. Therefore, we only use data from attribute manipulation to construct artifact masks. In fact, another recent work, FAKESHIELD [6] (which supports Image-Level Localization), also only supports forgery localization for the attribute manipulation task and abandons other forgery types.

---

> ### Author Response · Authors · 2025-11-16
> **For Other Weaknesses**
>
> ### For Weakness 2
> The core objective of our research differs from that of the CVPR 2025 paper "Towards General Visual-Linguistic Face Forgery Detection". The latter focuses more on enhancing the model’s forgery detection performance, whereas we aim to ground the concept of Artifact-Existing regions in forgery explanations to segmentation masks. Therefore, we provide more detailed annotations not to "translate to meaningful performance gains," but to meet the needs of data construction for the Artifact-Grounding Explanation (AGE) task. In fact, our work is indeed the first to explore the Artifact-Grounding Explanation (AGE) task.
>
> ### For Weakness 3
> We will explain why we did not report the performance of our FiFa-MLLM on previously established deepfake detection datasets.
> Our work is the first to explore the Artifact-Grounding Explanation (AGE) task, aiming to ground the concept of Artifact-Existing regions in forgery explanations to segmentation masks. We pursue the interpretability of forgery traces at the human perception level, requiring forgery localization consistent with human common sense. Specifically, it involves identifying the most unnatural regions in images, describing them in text, and providing corresponding mask annotations.
> This heightened pursuit of interpretability means we need to balance the performance of forgery Detection, Classification, Localization, Text-Only Explanation, and Artifact-Grounding Explanation during training, rather than deliberately optimizing the performance of forgery detection as a binary classification task.
> Thus, it would be unfair to directly compare our method with existing approaches on previously established binary classification datasets (e.g., FaceForensics++, Celeb-DF, and Wild DeepFake). However, on DD-VQA and DFA-Bench, constructed for Explainable DeepFake Analysis, we have achieved SOTA results on the forgery detection task, demonstrating the potential of our model.
> In the future, we will try to obtain more computational resources to explore strategies such as extending training steps or adding other supervision losses to pursue higher performance in both forgery detection and interpretability metrics simultaneously.
>
>
> ### For Weakness 4
> Our method achieves particularly remarkable gains (not merely marginal improvements) on the Text-Only Explanation task of DD-VQA, with the following performance metrics: BLEU-4 increased from 40.8 to 48.1, CIDEr from 2.057 to 2.869, ROUGE-L from 60.9 to 65.4, and METEOR from 34.6 to 40.4.
>
> Regarding the latest XDFA method, FFAA has not yet made its data publicly available to date. Constrained by computational resources (i.e., funding constraints), we are currently unable to provide performance evaluations on the FakeShield baseline within a short timeframe. Additionally, FakeShield is not a work specifically designed for explainable face forgery analysis.

---

> ### Author Response · Authors · 2025-11-16
> **For Questions**
>
> ### For Question 1
> Our research objective is to construct a data annotation pipeline supporting the Artifact-Grounding Explanation (AGE) task, rather than improving the performance of DeepFake Detection by providing more detailed annotations. Thus, there is no need to introduce additional ablation experiments to compare detection performance under different concept granularities of FICT. The granularity of 112 concepts was determined in practice to facilitate the data construction of the AGE task.
>
> ### For Question 2
> The performance improvements stem from face-specific pre-training and unfreezing the visual encoder for training, rather than from better training data or unified encoder design. The detailed analysis is as follows:
>
> In all model configurations presented in Table 6, training is conducted on our training data with identical hyperparameter settings. This allows us to rule out the possibility that performance gains originate from better training data.
>
> Regarding the unified encoder design, we do not claim that the unified encoder design itself can significantly enhance model performance. We argue that in the Explainable DeepFake Analysis task, this design can significantly reduce parameters, improve operational efficiency, and simultaneously achieve performance exceeding the strong baseline GLaMM. In our ablation experiments, we followed the settings in GLaMM: we kept the global image encoder CLIP and grounding image encoder SAM frozen. Subsequently, we replaced the CLIP+SAM encoders with the FaRL encoder (also kept frozen), achieving results that outperformed CLIP+SAM on the Det., Cls., and Loc. tasks, while obtaining comparable results on the TOE and AGE tasks. This demonstrates the potential of FaRL as a visual encoder replacement for CLIP+SAM. Notably, FaRL reduces parameters by 0.94B compared to CLIP+SAM, and the unified encoder avoids repeated encoding of the same image during training and inference, significantly improving application efficiency.
>
> Subsequently, we further unfroze FaRL's parameters for training to further enhance model performance—a performance improvement at this step that aligns with general expectations.
>
> In summary, the performance gains primarily derive from FaRL's pre-training (which enables a strong understanding of faces) and unfreezing FaRL's parameters for training. Our primary goal is to achieve performance on par with or exceeding the GLaMM baseline while improving application efficiency. In our experiments, we have successfully achieved this goal.
>
> ### For Question 3
> We believe that after introducing region mask prediction, although the improvement margin is relatively small, it still suffices to demonstrate the effectiveness of region mask prediction.
> Regarding TOE METEOR, there appears to be an inaccuracy in the changes you noted. After introducing regional mask prediction, TOE METEOR increased from 21.3 to 21.4, with no negative impact incurred.

---

> ### Author Response · Authors · 2025-11-16
> **References**
>
> [1] Nguyen, H. H., Fang, F., Yamagishi, J., & Echizen, I. (2019, September). Multi-task learning for detecting and segmenting manipulated facial images and videos. In 2019 IEEE 10th international conference on biometrics theory, applications and systems (BTAS) (pp. 1-8). IEEE.
>
> [2] Kong, C., Chen, B., Li, H., Wang, S., Rocha, A., & Kwong, S. (2022). Detect and locate: Exposing face manipulation by semantic-and noise-level telltales. IEEE Transactions on Information Forensics and Security, 17, 1741-1756.
>
> [3] Lai, Y., Luo, Z., & Yu, Z. (2023, December). Detect any deepfakes: Segment anything meets face forgery detection and localization. In Chinese conference on biometric recognition (pp. 180-190). Singapore: Springer Nature Singapore.
>
> [4] Li, L., Bao, J., Zhang, T., Yang, H., Chen, D., Wen, F., & Guo, B. (2020). Face x-ray for more general face forgery detection. In Proceedings of the IEEE/CVF conference on computer vision and pattern recognition (pp. 5001-5010).
>
> [5] Dang H, Liu F, Stehouwer J, et al. On the detection of digital face manipulation[C]//Proceedings of the IEEE/CVF Conference on Computer Vision and Pattern recognition. 2020: 5781-5790.
>
> [6] Xu, Z., Zhang, X., Li, R., Tang, Z., Huang, Q., & Zhang, J. FakeShield: Explainable Image Forgery Detection and Localization via Multi-modal Large Language Models. In The Thirteenth International Conference on Learning Representations.

---

### Official Review · Reviewer_8RAv · 2025-11-01

**Soundness:** 1
**Presentation:** 3
**Contribution:** 1
**Rating:** 2
**Confidence:** 5

**Summary:**

The paper introduces a benchmark that moves beyond simple detection to fine-grained, explainations. The authors propose an automated pipeline, FiFa-Annotator, to generate a large-scale dataset where forged facial images are annotated with detailed textual explanations and corresponding segmentation masks that pinpoint the manipulated regions. They also introduce FiFa-MLLM, a unified multimodal model designed to perform a new "Artifact-Grounding Explanation" task, which involves generating natural language descriptions of forgeries that are visually grounded by these masks.

**Strengths:**

- The paper makes a commendable attempt to advance DeepFake analysis beyond simple binary real vs. fake detection to fine-grained, explainable localization and description.
- The proposed FiFa-MLLM architecture is a thoughtful effort to create a unified, multi-task model that can handle diverse inputs like bounding box queries.

**Weaknesses:**

- FiFa is designed specifically for DeepFakes created using "attribute manipulation" techniques. The authors state that this is because other methods, like identity or expression swapping, create pixel-level changes that are not localized to the artifact, making their artifact detection method unreliable. This limits the training data for fine-grained explanation to a single class of forgery. In fact there are several methods in the literature that can detect and localize identity or expression swaps. So, if existing methods can detect and/or localize them, making these annotations a part of the benchmark is crucial.
- The annotations are generated using proprietary models like GPT-4o. This assumes that GPT-4o is a good DeepFake detector and reasoning model, which is wrong. MLLMs are inherently built for global semantics rather than capturing subtle inconsistencies that occur in DeepFakes. Papers like [1, 2] show that subtle changes are ignored, and these models are broadly good in overall content understanding. This means that the annotations are highly unreliable and noisy.
- Artifact masks are created by identifying the top 5% of pixel intensity differences between a real and fake image. This heuristic may not be robust enough to capture subtle manipulations or forgery types that involve smoother, less distinct alterations. The choice of a fixed 5% threshold is also not justified and may be suboptimal. If pixel intensity differences were enough for DeepFake detections, there would not be a need for complicated pipelines for DeepFake detection that are present in the literature.
- Since the FiFa-MLLM is trained and evaluated on a benchmark (FiFa-Bench) created by its own FiFa-Annotator pipeline, there is a risk that the model is learning the specific patterns of the annotation process rather than generalizable features of DeepFakes, through a "shortcut" learning method. There are no qualitative results that show that the generated explanations are actually grounded in visual cues.

[1] Tong, Shengbang, et al. "Eyes wide shut? exploring the visual shortcomings of multimodal llms." Proceedings of the IEEE/CVF Conference on Computer Vision and Pattern Recognition. 2024.
[2] Huynh, Ngoc Dung, et al. "Vision-Language Models Can't See the Obvious." Proceedings of the IEEE/CVF International Conference on Computer Vision. 2025.

**Questions:**

- Why was a fixed 5% threshold chosen for creating artifact masks? Were alternative, more adaptive methods for identifying manipulated regions explored? And why does it make sense to flag artifacts solely based on this concept of pixel intensity?
- What specific methods were used to verify the factual accuracy of the forgery descriptions generated by GPT-4o? How was the claim of achieving "fewer hallucinations" quantified?
- How well does the model perform its primary AGE task on forgeries it was not trained on for this task, such as identity swaps, expression swaps, or entire face synthesis, i,e, in cross-data settings?
- The addition of auxiliary supervision for region masks improved localization but slightly hurt text generation scores. Could the authors elaborate on this potential trade-off between visual grounding accuracy and textual explanation quality?

---

> ### Author Response · Authors · 2025-11-16
> **For Weakness 1**
>
> Thank you for your insightful questions. We feel there may be some subtle gaps in understanding regarding the core objective of our paper and the current technical landscape in the field of forgery localization, and we will provide a detailed response to your concerns in the following.
>
> ### For Weakness 1
> Our work has incorporated various forgery techniques for the forgery detection task.
> As shown in the first row of Table 2, we used data from FFHQ, CelebA, and DFFD in our constructed training and test sets for the DeepFake Detection task. The DFFD dataset covers four types of forgery techniques: identity swapping, expression swapping, attribute manipulation, and entire face synthesis samples. This point has been illustrated in Section 3.3 of the main text and Appendix A.4.
>
> Regarding the incorporation of multiple forgery techniques in the forgery localization task, we believe the existing literature lacks sufficient referable solutions. The specific analysis is as follows:
> Currently, existing forgery localization techniques adopt two methods for constructing forgery masks:
> 1. Treating all pixels or boundary pixels of the entire forged facial region as the forgery mask, as in References [1]-[4]. The motivation behind these methods is that some early face forgery techniques cut out the facial region to be forged and replace it with a newly generated one. These forgery methods are "whole-face level" and clearly cannot meet the needs of fine-grained face forgery localization in our work.
> 2. Calculating pixel intensity differences between real images and forged images followed by binarization to obtain the forgery mask, as in References [5]-[6]. This method is also adopted in our work.
>
> It should be noted that existing non-MLLM-based forgery localization works mainly serve as auxiliary tasks for forgery detection to improve the performance of binary classification in forgery detection. Therefore, they do not care whether the localized forged regions are consistent with human perception.
>
> Our work is the first to explore the Artifact-Grounding Explanation (AGE) task, aiming to ground the concept of Artifact-Existing regions in forgery explanations to segmentation masks. We pursue the interpretability of forgery traces at the human perception level, requiring forgery localization consistent with human common sense.
>
> From an empirical perspective, attribute manipulation concentrates unnatural traces in the altered regions. You can refer to the image samples presented in the main text and the appendix. It can be observed that in the FaceAPP data we used, the regions with the largest differences in the forgery masks are visually consistent with human common sense of forgery.
>
> However, this is not the case for forged images generated through identity swapping or expression swapping. Due to significant changes in identity and expression, the positions of facial concepts in forged images shift significantly from those in original images. Consider the following scenario: in identity swapping, the eyes of the source image are relatively above those of the target image, leading to the eyes in the forged image also being above those in the original real image. Calculating pixel differences at this time will leave traces of two pairs of eyes (because they are misaligned) in the forgery mask. Nevertheless, we cannot confirm that these regions with the largest differences in the forgery mask are necessarily the most visually unnatural. A similar issue exists in expression swapping.
>
> We acknowledge that relying solely on attribute manipulation as the source of data for forgery localization and the AGE task limits the capabilities of our model. However, based on our investigation, there is currently no more effective method to obtain forgery localization masks consistent with human common sense automatically. As an exploratory work, we aim to prioritize ensuring data quality rather than hastily attempting to cover all four types of forgery techniques in the absence of a mature automated solution. Therefore, we only use data from attribute manipulation to construct artifact masks. In fact, another recent work, FAKESHIELD [6] (which supports Image-Level Localization), also only supports forgery localization for the attribute manipulation task and abandons other forgery types.

---

> ### Author Response · Authors · 2025-11-16
> **For Weakness 2 and Question 2**
>
> ### For Weakness 2 and Question 2
> The purpose of this paper is to align forgery explanations with segmentation masks of manipulated artifacts, thereby providing fine-grained forgery localization that is consistent with human common sense. We DO NOT require GPT-4o to independently discover regions with forgery traces. Instead, we first compute the forgery masks, determine the Artifact-Existing Concepts following the approach outlined in Step 2 of Section 3.2, and then use this list of Artifact-Existing Concepts as a constraint to guide GPT-4o to focus on these specific regions. Within each region, we also DO NOT expect GPT-4o to detect subtle inconsistencies; rather, it only needs to understand the visually observable content.
>
> In fact, we have improved upon existing data annotation methods in the field of Explainable DeepFake Analysis. Existing works [6-8] generally input original images and forged images directly, requiring GPT-4o to independently discover and analyze forgery traces to construct training and evaluation data.
>
> In contrast, by incorporating prior knowledge, we directly identify regions containing forgeries rather than delegating this determination to GPT-4o, thereby ensuring the factual accuracy of the forgery descriptions. Next, we will quantify the "fewer hallucinations" in our annotation pipeline.
>
> As noted in Section 5 of the paper, for fair comparison, we constructed FaceAPP-VQA using an automated annotation pipeline similar to existing approaches, and we also provided the prompt used to construct FaceAPP-VQA in Table 15. This prompt instructs GPT-4 to determine regions containing forgeries. We randomly extracted 100 samples with identical image IDs from both FaceAPP-VQA and the I-AGE task subset of FiFa-Instruct-1M and manually inspected whether the regional concepts included in their textual explanations are consistent with the artifact masks.
>
> Upon inspection, the 100 samples from FiFa-Instruct-1M contained a total of 889 regional concepts, with artifact masks present for all corresponding regions. In contrast, the 100 samples from FaceAPP-VQA contained 612 regional concepts, among which only 383 had actual artifact masks—accounting for 62.6% of the total. This confirms that our annotation pipeline indeed exhibits "fewer hallucinations."

---

> ### Author Response · Authors · 2025-11-16
> **For Other Weaknesses and Questions**
>
> ### For Weakness 3 and Question 1
> As detailed in our response to Weakness 1, using pixel intensity differences to localize forgery masks is a common and reasonable practice in current data processing for Forgery Localization and Explainable DeepFake Analysis.
> This does not imply that "pixel intensity differences were enough for DeepFake detections." In fact, the purposes of these two tasks are fundamentally different. For forgery detection, the focus is on the model’s end-to-end binary classification performance, without concern for the specific clues it relies on. In contrast, for fine-grained Explainable DeepFake Analysis, we aim to align forgery explanations with segmentation masks of manipulated artifacts, so we need pixel intensity differences to construct the ground truth for artifact masks.
>
> Based on empirical evidence, we set the threshold at the top 5%. The resulting Artifact Masks can filter out discontinuous small-area masks—given that the input images in our scenario are 299×299 pixels, masks with fewer than 50 pixels are considered small-area masks.
> In some existing methods [5], fixed pixel difference thresholds are even employed to extract masks. In this paper, we instead adopt a relative threshold, selecting the top 5% to extract masks, which already exhibits a certain degree of adaptability.
>
> ### For Weakness 4
> We provide numerous visualization examples in Appendix C, which can qualitatively demonstrate that the regions with artifacts referred to in the generated explanatory texts can align with the locations perceived by humans as forgery traces in the images.
>
> ### For Question 3
> Due to the reasons described in our response to Weakness 1, there is currently a lack of appropriate techniques to extract forgery localization masks consistent with human common sense for images generated by techniques such as identity swapping, expression swapping, or entire face synthesis. Consequently, we are unable to test the performance under these scenarios.
>
> ### For Question 4
> Constrained by computational resources, we were unable to conduct a thorough search for the optimal weight settings of the auxiliary supervision task loss functions. In our experiments, we strived to maintain the text generation scores roughly unchanged (in TOE, METEOR slightly increased from 21.3 to 21.4; in AGE, METEOR slightly decreased from 21.7 to 21.6) while evaluating the performance of forgery mask prediction. From the results, the mIoU in the Loc. task increased from 30.0 to 30.9, and in the AGE task from 30.0 to 30.4. These outcomes sufficiently demonstrate the effectiveness of introducing regional mask auxiliary supervision in enhancing mask prediction performance.

---

> ### Author Response · Authors · 2025-11-16
> **References**
>
> [1] Nguyen, H. H., Fang, F., Yamagishi, J., & Echizen, I. (2019, September). Multi-task learning for detecting and segmenting manipulated facial images and videos. In 2019 IEEE 10th international conference on biometrics theory, applications and systems (BTAS) (pp. 1-8). IEEE.
>
> [2] Kong, C., Chen, B., Li, H., Wang, S., Rocha, A., & Kwong, S. (2022). Detect and locate: Exposing face manipulation by semantic-and noise-level telltales. IEEE Transactions on Information Forensics and Security, 17, 1741-1756.
>
> [3] Lai, Y., Luo, Z., & Yu, Z. (2023, December). Detect any deepfakes: Segment anything meets face forgery detection and localization. In Chinese conference on biometric recognition (pp. 180-190). Singapore: Springer Nature Singapore.
>
> [4] Li, L., Bao, J., Zhang, T., Yang, H., Chen, D., Wen, F., & Guo, B. (2020). Face x-ray for more general face forgery detection. In Proceedings of the IEEE/CVF conference on computer vision and pattern recognition (pp. 5001-5010).
>
> [5] Dang H, Liu F, Stehouwer J, et al. On the detection of digital face manipulation[C]//Proceedings of the IEEE/CVF Conference on Computer Vision and Pattern recognition. 2020: 5781-5790.
>
> [6] Xu, Z., Zhang, X., Li, R., Tang, Z., Huang, Q., & Zhang, J. FakeShield: Explainable Image Forgery Detection and Localization via Multi-modal Large Language Models. In The Thirteenth International Conference on Learning Representations.
>
> [7] Huang, Z., Xia, B., Lin, Z., Mou, Z., Yang, W., & Jia, J. (2024). Ffaa: Multimodal large language model based explainable open-world face forgery analysis assistant. arXiv preprint arXiv:2408.10072.
>
> [8] Qin, L., Jiang, N., Zhang, Y., Qiu, Y., Zeng, D., Hu, J., & Deng, W. (2025, April). Towards Interactive Deepfake Analysis. In ICASSP 2025-2025 IEEE International Conference on Acoustics, Speech and Signal Processing (ICASSP) (pp. 1-5). IEEE.

---

### Meta-Review · Area_Chair_oXqA · 2026-01-05

**Summary:**

The reviewers highlighted the granularity of explanation sought by the proposed method and the variety of tasks addressed by the proposed FiFa-MLLM. In a similar manner, the reviewers found the proposed annotation framework a proper contribution.

During the discussion some initial doubts/concerns were clarified. Moreover, qualitative result, missing in the original submission were added under request.

There were concerns regarding the completeness of the conducted evaluation which is an obstacle towards assessing the generality of the proposed method/framework and reported results. Good part of the response provided to these concerns hinted at targeted pursuit of interpretability of forgery traces at the human perception level on 11 tasks and that further improvements were not pursuit due to limited time and computational resources. In this regard, perhaps it would have been more desirable to strike a better balance between the number of targeted capabilities/tasks and the depth of the evaluation/analysis for the selected capabilities/tasks. From the discussion it seems that depth of the analysis seems to have been sacrificed in some parts.

Related to the previous point, in several parts of the rebuttal it is reiterated the pursued goal of interpretability at the human perception level. Counterintuitively, there is no user study to assess the level to which this goal is successfully achieved.

Beyond that, there were several concerns raised regarding the bias introduced by the MLLMs in the generation of explanations and the practice of training and evaluating from very related sources. While some of these concerns were addressed in the rebuttal, the fact that every single reviewer raised a related concern signals issues with clarity of the presented content.

Finally, some of the targeted explainability goals, fine-grained, multimodal and more localized, while perhaps novel within the deepfake detection application, they are not new considering the larger umbrella of AI explainability (see below for some examples). The manuscript would benefit from positioning the proposed components in that context.

References

- Xiao Rui et al., "FLAIR: VLM with Fine-grained Language-informed Image Representations", CVPR 2025.

- Sammani Fawaz et al., "NLX-GPT: A Model for Natural Language Explanations in Vision and Vision-Language Tasks", CVPR 2022

- Hendricks LA, et al. "Generating visual explanations with natural language". Applied AI Letters. 2021; 2(4):e55. doi:10.1002/ail2.55

**Reviewer Concerns:**

Addressed Concerns:

- Reviewer 8RAv

    - Unreliability of synthetic annotations produced via MLLMs.

    - Strong assumptions on pixel-intensity characteristics of manipulated image regions and related heuristic solution.

- Reviewer iCL8

    - Not applicable

- Reviewer nV3V

    - Limited amount of qualitative results.

- Reviewer Vjsn

    - Limited evaluation on the alignment of linguistic and visual outputs.

    - Missing qualitative results.


Outstanding Concerns:

- Reviewer 8RAv

    - Limited scope; only focus on deepfakes created via attribute manipulation techniques.

    - Since the proposed FiFa-MLLM is trained and evaluated on annotations produced by the proposed FiFa-Bench benchmark the model might not be able to learn more general patterns.

- Reviewer iCL8

    - Limited scope; only focus on deepfakes created via attribute manipulation techniques.

    - Missing (quantitative) comparison w.r.t. similar existing efforts.

    - Limited evaluation: focused on its own benchmark (FiFa-Bench) and include very few existing efforts (GLaMM). No evaluation on out-of-distribution scenarios nor real-world deployment scenarios.

    - Relatively modest performance improvements and quantitative comparison w.r.t. more recent XDFA methods.

- Reviewer nV3V

    - Bias injected in the explanations by the used MLLMs.

- Reviewer Vjsn

    - No experiments on standard benchmarks and including  existing related methods. Hard to assess the generality of the reported results/observations.

    - Was the output of the proposed method rated by humans?

**Reviewer Scores:**

The paper received average reviews. While a relatively extended rebuttal was provided to each review, due to the similar concerns that were raised, there was some repetition in the provided responses. Some of these responses were not sufficiently conclusive as to properly address some of the major concerns, i.e. the limitations on the evaluation (and generality of the reported results), and a missing assessment on the level to which the provided explanations were at the human level. Given these grounds, I find unlikely that the reviewers would have updated their initial scores in a significant degree as to warrant the acceptance of the paper.

---

### Decision · Program_Chairs · 2026-01-26

Reject